

# Impact of AMV on rainfall intensity distribution and timing of the West African Monsoon in DCPP-C-like simulations

Elsa Mohino[1], Paul-Arthur Monerie[2], Juliette Mignot[3], Moussa Diakhaté[4], Markus Donnat[5,6], Christopher David Roberts[7], and Francisco Doblas-Reyes[5,6]

[1]Physics of the Earth and Astrophysics Department, Complutense University of Madrid, 28040 Madrid, Spain
[2]National Centre for Atmospheric Science, Reading, United Kingdom
[3]LOCEAN/IPSL, IRD/Sorbonne Université/CNRS/MNHN, 4 place Jussieu, 75005 Paris, France
[4]École Supérieure des Sciences et Techniques de l'Ingénieur, Université Amadou Mahtar Mbow, rue 20x21 Pôle Urbain de Diamniadio, 20000 Dakar, Sénégal
[5]Barcelona Supercomputing Center (BSC), 08034 Barcelona, Spain
[6]Institució Catalana de Recerca i Estudis Avançats (ICREA), 08010 Barcelona, Spain
[7]ECMWF, Shinfield Park, Reading, RG2 9AX, United Kingdom

**Correspondence:** Elsa Mohino (emohino@ucm.es)

**Abstract.** Previous studies agree on an impact of the Atlantic Multidecadal Variability (AMV) on total seasonal rainfall amounts over the Sahel. However, whether and how AMV affects the distribution of rainfall or the timing of the West African Monsoon is not well known. Here we analyze daily rainfall outputs from atmosphere-ocean coupled models. Models show dry biases over the Sahel, where the mean intensity is consistently smaller than observations, and wet biases over the Guinea Coast, where they simulate too many rainy days. In addition, most models underestimate the average length of the rainy season over the Sahel, some due to a too late monsoon onset and others due to a too early cessation. In response to a persistent positive AMV pattern imposed in the Atlantic, following a protocol largely consistent with the one proposed by the Component C of the Decadal Climate Prediction Project (DCPP-C), models show an enhancement in total summer rainfall over West African land mass, including the Sahel. Both the number of wet days and the intensity of daily rainfall events are enhanced over the Sahel. The former explains most of the changes in seasonal rainfall in the northern fringe, while the latter is more relevant in the southern region, where higher rainfall anomalies occur. This dominance is connected to the changes in the number of days per type of event: the frequency of both moderate and heavy events increases over the Sahel's northern fringe. Conversely, over the southern limit, it is mostly the frequency of heavy events which is enhanced, affecting the mean rainfall intensity there. Extreme rainfall events are also enhanced over the whole Sahel in response to a positive phase of the AMV. Models with stronger negative biases in rainfall amounts tend to show weaker changes in response to AMV, suggesting systematic biases could affect the simulated responses. The monsoon onset over the Sahel shows no clear response to AMV, while the demise tends to be delayed and the overall length of the monsoon season enhanced between 2 and 5 days with the positive AMV pattern. The effect of AMV on the seasonality of the monsoon is more consistent to the West of 10ºW, with all models showing a statistically significant earlier onset, later demise and enhanced monsoon season with the positive phase of the AMV. Our results suggest a potential for the decadal prediction of changes in the intraseasonal characteristics of rainfall over the Sahel, including the occurrence of extreme events.



# 1 Introduction

The Atlantic Multidecadal Variability (AMV) is a basin-scale fluctuation at multi-decadal timescales observed in the Atlantic Sea Surface Temperature (SST) with high spatial coherence (e.g., Zhang et al., 2019). Its positive phase consists in anomalously

warm SST over the North Atlantic, while the negative one presents an anomalous cooling (Fig. 1a). The swings between positive and negative phases observed during the Twentieth Century take around 30-40 years (Kerr, 2000, Fig1b), though its spectrum shows a broad band of low-frequency signals (Zhang, 2017).

There is currently a debate on the origin of the AMV. Some works suggest fluctuations of the Atlantic Meridional Overturning Circulation as its main cause, highlighting the relevance of variability internal to the climate system (Knight et al., 2005;

Zhang, 2017; Kim et al., 2018; Zhang et al., 2019; Baek et al., 2022). On the other hand, other studies suggest a prominent role of changes in the aerosol atmospheric burden as an explanation for the observed AMV during the instrumental period, highlighting external forcings either from natural or anthropogenic aerosol sources (Rotstayn and Lohmann, 2002; Ottera et al., 2010; Booth et al., 2012; Terray, 2012; Watanabe and Tatebe, 2019). There are also studies suggesting that a combination of both internal variability and external forcings shapes the observed AMV (Terray, 2012; Qin et al., 2020).

Regardless of its origin, there is high consensus on the broad impacts of the AMV (interested readers are referred to Zhang et al., 2019, for a detailed review). It has been shown to modulate the location of the Atlantic Intertropical Convergence Zone (ITCZ), promoting in its positive phase a northward shift of the ITCZ and enhanced rainfall over Amazonia, decreased rainfall over Brazil's Nordeste and an increased frequency of Atlantic hurricanes (e.g. Knight et al., 2006; Trenberth and Shea, 2006; Zhang and Delworth, 2006; Villamayor et al., 2018a; Hodson et al., 2022). Observed and simulated results also suggest that it

is positively associated with summer surface warming and negatively associated with sea level pressure anomalies over North America and Europe (Sutton and Hodson, 2005; Knight et al., 2006; Qasmi et al., 2020). Away from the Atlantic, AMV can also promote wetter than average conditions for the Indian Monsoon, warmer than average conditions over northeast Asia, a cooling over the eastern and central tropical Pacific, modifying its inter-annual variability (Ruprich-Robert et al., 2017, 2021; Monerie et al., 2019, 2021; Hodson et al., 2022). The AMV has also been suggested as a possible modulator of Atlantic-Pacific

inter-basin connections at inter-annual time scales (Martín-Rey et al., 2015, 2018).

There is also a broad consensus, based on observations and modeling studies, that AMV modulates the West African Monsoon by promoting enhanced summer seasonal rainfall over the semi-arid area of the Sahel (e.g. Folland et al., 1986; Knight et al., 2006; Zhang and Delworth, 2006; Mohino et al., 2011; Ting et al., 2011; Martin and Thorncroft, 2014; Martin et al., 2014; Villamayor et al., 2018b; Monerie et al., 2019; Hodson et al., 2022) (Fig. 1c). However relevant, these findings do not clarify

whether and how AMV might affect the intra-seasonal characteristics of West African summer rainfall. The intra-seasonal characteristics, such as the timing of the monsoon season, the rainfall frequency and intensity and the frequency of different types of events, are particularly important for agricultural planning, especially over the Sahel (Ingram et al., 2002; Sultan et al., 2005; Guan et al., 2015).

Studies have evidenced changes in some intra-seasonal characteristics of Sahel rainfall at decadal and longer timescales.

Extreme rainfall events over the Sahel have been shown to strongly increase in recent decades (e.g. Ly et al., 2013; Panthou



et al., 2014, 2018; Sanogo et al., 2015; Taylor et al., 2017; Blanchet et al., 2018; Diatta et al., 2020; Chagnaud et al., 2022), while trends in extremes for previous periods were less consistent (Ly et al., 2013; De Longueville et al., 2016; Panthou et al., 2013). Results are also contrasting regarding the number of rainy days, with some works suggesting an enhancement in recent decades (Sanogo et al., 2015), while others suggesting a stability (Panthou et al., 2014, 2018). However, it is not clear to what extent these changes could be related to AMV.

Based on daily gauge records in the 1918-2000 period, Badji et al. (2022) suggest that indeed AMV could be modulating some intra-seasonal characteristics of rainfall, including extreme event occurrence, at least over Senegal, located at the westernmost Sahel. However, the unequivocal attribution of observed changes to AMV is hindered by the presence of other sources of long-term SST variability affecting the Sahel (Mohino et al., 2011). In addition, the recent decoupling of east and west Sahel rainfall recovery since the drought of the 1970-80s (Lebel and Ali, 2009), which also appears in the trends in the extremes (Blanchet et al., 2018; Panthou et al., 2018), calls for attention when extrapolating the results obtained regionally to the whole Sahel. The shortness of the observed record, with less than two full AMV cycles since the beginning of the Twentieth Century, also hinders the robust estimation of AMV impacts.

Using model simulations provides an additional pathway to evaluate the influence of AMV on West African rainfall considering other characteristics beyond the seasonal amounts. The modeling approach provides a way to clearly separate the effects of the observed AMV pattern from other sources of long-term variability and allows robust estimations by simulating a large number of ensemble members and longer time spans than the observational short record. Here we take this approach and analyze the output of experiments run with global coupled climate models in which the North Atlantic SSTs are flux-corrected to provide idealized AMV-like anomalies, as proposed in component C of the Decadal Climate Prediction Project (DCPP-C) (Boer et al., 2016). Our main aim is to evaluate the impact of the observed AMV pattern on intraseasonal characteristics of rainfall over West Africa. We particularly focus on five intraseasonal characteristics: the number of rainy days, the mean intensity of rainfall, the occurrence of moderate, heavy and extreme rainfall events, and the timing of the monsoon season. We also compare the simulations from different models run under a largely common protocol to provide an additional measure of the robustness of the results.

## 2 Data and methods

### 2.1 Observation-related products used

Three different rainfall datasets are used. To estimate model biases in the representation of rainfall and some of its intraseasonal characteristics, we use GloH2O's Multi-Source Weighted-Ensemble Precipitation version 2.8 (hereinafter MSWEP). It is a globally gridded rainfall data set at a horizontal resolution of 0.1° and temporal resolution of up to 3 hours spanning from 1979 to present. It is derived by merging satellite, reanalysis and rain gauge data (Beck et al., 2019). MSWEP has been shown to provide the most accurate estimates at daily resolution over West Africa when compared to other 22 daily rainfall products (Satgé et al., 2020). To estimate the biases, we average the MSWEP dataset over its whole "historical" period (1979-2020) of gauge-corrected data so as to avoid merging with slightly differently processed near-real-time values.



Gridded rainfall estimates are subjected to observational uncertainties coming from different sources (Turco et al., 2020).
To account for these uncertainties, we also use the Climate Hazards group Infrared Precipitation with Stations version 2.0 (hereinafter CHIRPS). It provides daily gridded rainfall from 1981 to present at a resolution of 0.05° over land between 50ºS and 50ºN by blending satellite and gauge estimates (Funk et al., 2015). It shows high performance over West Africa at monthly time scales, though it is outperformed by MSWEP at daily timescales (Funk et al., 2015; Satgé et al., 2020). Here we use the period 1981-2021 to calculate CHIRPS climatological values and compare them with MSWEP ones.

Lastly, monthly estimates of total rainfall and number of wet days from the Climate Research Unit TS v.4.06 dataset (hereinafter CRU) are used to estimate the AMV impact in observations. They are based on station observations and provided globally over land with a resolution of 0.5° and span the period from 1901 to present (Harris et al., 2020). The period used in this work is 1901 to 2013, which is the maximum possible coincident with the AMV index used to provide the boundary conditions for the sensitivity experiments (Boer et al., 2016). In addition, SSTs from the Extended Reconstructed SST dataset
version 4 (hereinafter referred to ERSSTv4) are used to estimate the observed AMV pattern. It is a global reconstruction of monthly SSTs provided at a resolution of 2º from 1857 to present (Huang et al., 2015). For consistency with the CRU dataset used to estimate the observed AMV impacts on West African rainfall, in this work we use data from 1901 to 2013.

## 2.2    Simulations

In the framework of the Coupled Intermodel Comparison Project phase 6 (CMIP6 Eyring et al., 2016), the Decadal Climate
Prediction Project (DCPP) has coordinated a series of experiments aimed at understanding the variability at decadal timescales (component C) (Boer et al., 2016). Here we make use of the AMV experiments in DCPP-C which are 10 year long coupled simulations in which the model's SST are restored in the North Atlantic basin, excluding the Mediterranean Sea, to follow a fixed and idealized pattern of SST anomalies representative of the observed AMV (Boer et al., 2016). The pattern is obtained as the regression of the SSTs onto the standardized AMV index over the 1900-2013 period (Fig. 1b), which is added (sub-
tracted) to the model's pre-industrial climatological SSTs for the AMV+ experiment (AMV-) (see Technical Note 1 and 2 at https://www.wcrp-climate.org/dcp-overview for more detailed information). Here, we focus on the linear component and estimate the impact of AMV as the change in a given quantity between both experiments. We first calculate the mean along the 10 years of simulation and average all ensemble members for each experiment. We note that there is no clear drift in rainfall over the Sahel in the simulations (not shown) and that taking 10-year averages allows for a more statistically robust estimation of the
impact of AMV, which is already established in the first year of the runs (not shown). We then subtract the AMV- experiment to the AMV+ experiment, which corresponds to a total SST anomaly of twice the AMV regression pattern (Fig. 1a). Though we cannot rule out non-linear effects of AMV might be present in the simulations, there is no run with an imposed pattern of no anomalies in the North Atlantic basin to estimate them.

In addition to the DCPP-C experiments, in the framework of EU Horizon 2020 PRIMAVERA project, a similar protocol
has been applied to evaluate the dependence of the models' responses on resolution (Hodson et al., 2022). The PRIMAVERA experiments also impose a fixed and idealized anomalous SST pattern for 10 year long simulations. However, they differ from the DCPP-C protocol in that they impose twice the anomalous AMV patterns and that the model's background climatology



is consistent with mid-Twentieth Century radiative forcings. To ease comparison with the DCPP-C set, changes associated with AMV for these runs are presented as half the difference between the response to 2AMV+ minus 2AMV- experiments. We refer to these experiments as PRIMAVERA protocol. Note, however, that the direct comparison of results from DCPP-C and PRIMAVERA protocols might be hindered by non-linearities in both the response to a stronger anomalous pattern and the superposition of this pattern to a different climatological background.

As we aim at exploring the effect of AMV not only on seasonal rainfall amounts but also on its intraseasonal characteristics, here we focus on a subset of the models for which daily rainfall outputs were available (Table 1).

## 2.3 Metrics

A rainy or wet day is defined as one in which the total amount of rainfall is above 1 mm (Hartmann et al., 2013). This definition has been applied widely over West Africa (e.g. Sanogo et al., 2015; Diaconescu et al., 2015; De Longueville et al., 2016; Diakhate et al., 2019; Diatta et al., 2020; Badji et al., 2022) and is consistent with the World Meteorological Organization recommendations (WMO, 2009). This definition is thus used throughout the study for the simulations and MSWEP and CHIRPS datasets. Note, however, that it differs from the one used in the CRU dataset to obtain the number of wet days, for which the threshold is 0.1 mm (Harris et al., 2020). Here we count the number of wet days ($n$) in the July to September (JAS) summer season. This season represents the mature phase of the West African Monsoon, when rainfall is well developed in the Sahel (Thorncroft et al., 2011).

As an estimate of the mean intensity of rainfall ($I$), we use the ratio of the total amount of rainfall accumulated on wet days to the number of wet days in the JAS season, this is, the average rainfall amount per rainy day, also referred to as Simple Daily Intensity Index (e.g. WMO, 2009; Zhang et al., 2011; Herold et al., 2016).

For consistency with previous works and with the CRU monthly observations, we evaluate the total rainfall in JAS ($P$) by accumulating daily rainfall for the $N = 92$ days of the season, regardless of the type of day (whether wet or not). We express it as a mean value per day ($p$, in mm/day) and we relate it to the number of wet days ($n$) and in the mean intensity ($I$) by defining $P_0$ as the total amount of rain (in mm) fallen during non rainy days (< 1 mm):

$$p = \frac{P}{N} = \frac{1}{N}(P_0 + n \cdot I) \tag{1}$$

The changes in the total amount of rainfall expressed as an average value per day ($\Delta p$) between the experiments (AMV+ minus AMV-) can then be related to changes in the number of wet days ($\Delta n$) and in the intensity ($\Delta I$) by:

$$\Delta p = \frac{1}{N}(\Delta P_0 + \Delta n \cdot \bar{I} + \bar{n} \cdot \Delta I + \Delta n \cdot \Delta I) \tag{2}$$

where $\Delta P_0$ is the difference in the amount of rain (in mm) fallen during non rainy days (< 1mm) and $\bar{n}$ and $\bar{I}$ are the climatological number of days and intensity, respectively. These climatological values are estimated for each model as the mean of both types of experiment (AMV+ and AMV-). Neglecting the changes in the precipitation falling in non-rainy days and changes coming from cross product of anomalies (which are usually smaller than 5% over West Africa, not shown), we can



approximate the change in as:

$$\Delta p \approx \frac{1}{N}(\Delta n \cdot \bar{I} + \bar{n} \cdot \Delta I) \qquad (3)$$

where the first term in the right hand side is the part explained by changes in the number of rainy days with no changes in mean intensity and the second term is the part explained by changes in the mean intensity with no changes in the number of rainy days.

Rainy days in JAS have been binned in different categories according to the percentiles in the distribution. Moderate (heavy)
events are defined as those rainy days for which the amount of rainfall fallen on that day is below (above) the 75 percentile, while for extreme rainfall events the 95 percentile is used. The thresholds for the percentiles are calculated independently at each grid point and take into account the total number of rainy days available in JAS. For the simulations, the threshold calculation for each model includes all days from the full 10 years of simulation, all members and both AMV+ and AMV- experiments. Here we count the number of moderate, heavy and extreme rainfall days per JAS season. Note that this methodology, although
based on wet days percentile thresholds, calculates frequency indices. It corresponds to the official recommendation of the World Meteorological Organization (WMO, 2009) and provides results qualitatively consistent with all-day percentiles (Schär et al., 2016).

For the onset, demise and total length of the monsoon season, we follow Liebmann et al. (2012) and calculate them locally at each grid point allowing for an annual regime only (we do not consider biannual regimes). This methodology has already
been applied to the West African Monsoon for observations and model outputs (e.g. Liebmann et al., 2012; Diaconescu et al., 2015; Dunning et al., 2016, 2017). It consists in calculating for each calendar year the dates for the minimum and maximum of the daily cumulative rainfall anomaly which provide the onset and cessation dates for the season, respectively. The total length of the season is given by the difference between the cessation and onset dates. For the simulations, the calculation is performed separately for each year in each ensemble member. The daily rainfall anomaly is obtained as the rainfall for each day minus the
long-term climatological mean daily rainfall using all available years in the observations and all years and ensemble members in both experiments, AMV+ and AMV-, in the models.

For the sake of briefness, the metrics related to the monsoon season are averaged over the Sahel box (orange box in Fig. 1cd), which is taken in this work as the region 10ºW-10ºE and 10ºN-20ºN. To better compare with the results of Badji et al. (2022) obtained over Senegal and explore possible east-west differences, we also show these metrics for the westernmost Sahel
(purple box in Fig. 1cd). Some metrics are also presented over West African grid points, which are selected as those delimited by the box 17ºW-25ºE, 5ºN-22ºN excluding the grid points to the west of the line connecting the points 17ºW, 12ºN and 10ºW, 5ºN (blue box in Fig. 1cd).

## 2.4 Statistical significance

To test whether the change in a given quantity is statistically significant we apply the parametric test for differences of means
under independence (Wilks, 2019). The 10-year long simulations show a small negative drift in the energy imbalance at the top of the atmosphere, which is related to a positive drift in the outgoing longwave radiation, statistically significant for some



models (not shown). For this reason, we do not treat each year of each ensemble member as an independent realization. Instead, we average the 10 years of simulation and consider each member as an independent realization. The same procedure is followed with all the metrics shown in this paper. The total number of members taken into account for each model is shown in table 1.

## 3 Results

### 3.1 Representation of West African climatological rainfall and its intraseasonal characteristics by models

We begin by presenting the biases of the models, which are estimated by first averaging the AMV+ and AMV- experiments and then subtracting the observational estimate. All models show a dipole of north-to-south mean JAS rainfall bias over West Africa (Fig. 2). For the CNRM-CM6-1, there is an east-to-west bias superimposed to this pattern. All in all, the models provide too dry conditions over the Sahel ranging from deficits of 0.3 mm/day for the CNRM-CM6-1 model with the DCPP-C protocol to 1.8 mm/day for the EC-Earth3 in DCPP-C and ECMWF-IFS-LR in PRIMAVERA ones, which roughly represent between 8 and 60 % of average rainfall over the Sahel from MSWEP (Fig. 3ab). According to Satgé et al. (2020), at monthly timescales CHIRPS dataset presents smaller bias with respect to gauge observed rainfall than MSWEP. Fig. 2h shows that CHIRPS provides wetter conditions when compared to MSWEP (as do other datasets, not shown), so the above values could be underestimating rainfall dry biases over the Sahel by about 0.5 mm/day (Fig. 3ab). The north-to-south rainfall biases are consistent with the warm biases of SST simulated by all models in the southeastern tropical Atlantic, which reach values well over 2ºC (not shown). These warm biases are a prominent feature of the current generation of general circulation models (Richter and Tokinaga, 2020; Farneti et al., 2022) were also present in previous model generations like CMIP5 (Mohino et al., 2019; Farneti et al., 2022), and are related to a southward shift of the ITCZ over West Africa with reduced precipitation over the Sahel and enhanced precipitation over the Gulf of Guinea (Losada et al., 2010; Richter and Tokinaga, 2020). The comparison of atmosphere-only and coupled atmosphere-ocean simulations performed in Roberts et al. (2018) with the ECWMF-IFS-LR and ECMWF-IFS-HR models provides further evidence to suggest that the main factor explaining the north-to-south rainfall biases over West Africa come from biases in SSTs.

According to MSWEP, the number of rainy days in the JAS season (92 days in total) range between above 90 days over the western coast of Guinea to less than 10 in the northern fringe of the Sahel (see contours in Fig. 4). On average over the Sahel, it provides 37 rainy days per season. Satgé et al. (2020) show that MSWEP is the dataset most consistent with gauge rainfall daily records over the Sahel, precisely concerning the detection of rainy days. The CHIRPS dataset shows consistent results over the northern Sahel (Fig. 4h). However, the number of rainy days is smaller south of 15ºN, especially over the Gulf of Guinea region (Fig. 4h), despite the enhanced total seasonal amount of rainfall in CHIRPS with respect to MSWEP (Fig. 2h). The pattern of differences between CHIRPS and MSWEP is similar to the one obtained with two other daily rainfall datasets, the African Rainfall Climatology Version 2 - ARC-2 (Novella and Thiaw, 2013) and the Global Precipitation Climatology Project (GPCP) Daily Precipitation Analysis Climate Data Record version 1.3 (Adler et al., 2017) (not shown). On average, CHIRPS shows a reduction of 6 days in the number of rainy days over the Sahel with respect to MSWEP (Fig. 3b), which amounts to an underestimation of nearly 20%.





The pattern of model biases in the number of rainy days is consistent with the one in total seasonal rainfall, with an under-estimation in the northern Sahel and an overestimation towards the south in both magnitudes (Fig. 2 and 4 panels a-g), and a superposed east-to-west pattern the case of the CNRM-CM6-1 model. Taking CHIRPS differences with MSWEP as a measure of observational uncertainty, the models clearly overestimate the number of rainy days in the coastal region of Ivory Coast (Fig. 4). On average, over the Sahel, models present an underestimation in the number of rainy days per season ranging between

1 and 20 days for the IPSL-CM6–LR and ECMWF-IFS-LR models, respectively (Fig. 3b). However, only in the case of EC-Earth3, ECMWF-IFS-HR and ECMWF-IFS-LR models this underestimation is well beyond observational uncertainty. This suggests that, over the Sahel, the underestimation in seasonal rainfall amounts cannot be only related to the models' tendency to underestimate the number of rainy days.

       According to the MSWEP database, the mean intensity of rainfall falling on rainy days ranges from well above 12 mm/day

in the western coast, close to the Fouta Djallon and Guinea highlands, to below 4 mm/day north of 18ºN (see contours in Fig. 5). Rainfall mean intensity is much higher for CHIRPS over the whole West Africa, especially in the Guinea Coast with differences of over 12 mm/day, more than double the estimates obtained from MSWEP locally. Though both datasets show weaker differences in the mean intensity over the Sahel (nearly 4 mm/day on average, Fig. 3a), it still represents more than a 60% increase in CHIRPS with respect to MSWEP estimates. According to Satgé et al. (2020) and the previous results,

this difference between datasets could come from both an underestimation of rainy days in CHIRPS and an underestimation of rainfall amounts in MSWEP. In addition, the comparison of extreme rainfall from different datasets for Burkina Faso performed by Sanogo et al. (2022) suggests that MSWEP could be underestimating very extreme rainfall values, affecting its estimates of mean rainfall intensity.

       Once again, the bias pattern in the mean rainfall intensity in models is consistent with the one for total seasonal rainfall

(compare Figs. 2 and 5), with an underestimation of intensity over the Sahel (and to the east of West Africa in the CNRM-CM6-1 model) and an overestimation over the Guinea coast. However, looking at the differences between CHIRPS and MSWEP (Fig. 5h), the latter is well within observational uncertainty. This suggests that the overestimation of seasonal rainfall over the Gulf of Guinea could be more clearly linked to an overestimation of the number of rainy days simulated by models rather than an overestimation in the mean intensity of events. Regarding the Sahel region, most models underestimate the mean intensity

well beyond the observational uncertainty (Fig. 3a). The average underestimation with respect to MSWEP in this region ranges from 0.07 mm/day for the CNRM-CM6-1 model in the DCPP-C protocol (slightly above 1 % of the MSWEP estimate) to 1.7 mm/day for the EC-Earth3 (approximately 28 % of the MSWEP estimate).

       Even though the systematic negative bias in seasonal JAS rainfall shown by models over the Sahel is more consistent with a systematic underestimation of mean rainfall intensity when taking into account observational uncertainty as discussed above,

the spread in the models' simulated JAS seasonal rainfall seems to be better explained by the simulated number of rainy days (compare Fig. 3a and 3b): models that simulate higher amounts of total accumulated rainfall throughout the JAS season tend to be those models that simulate a higher number of rainy days, and also those that tend to simulate a higher intensity per rainy event, but the linear regression strength as measured by the correlation coefficient is higher in the first case. This could be connected to a higher disparity in the number of rainy days simulated by the models (a spread of roughly 70% taking into





account a range of 19 days for a multi-model mean of 27 days per season) than in the intensity (a spread of roughly 30 %, as measured by a range of 1.6 mm/day for a multi-model mean of 5.4 mm/day). The linear regression fit obtained across the models is more consistent with observational estimates from MSWEP than those from CHIRPS (compare interception of the regression fit with the observational lines). Note that these results should be taken with care, as the actual degrees of freedom are low due to the low number of models available with daily rainfall data and the redundancy of models, with similar models

under different resolutions, configurations, or protocols.

Despite the differences shown by MSWEP and CHIRPS datasets in the JAS total seasonal rainfall, number of rainy days and, especially, the mean rainfall intensity, they both agree on the average monsoon season timing over the Sahel (Fig. 6a). The average onset date is June 12th (13th), the average cessation date is September 20th (23rd), with a total monsoon season length of 100 (102) days for the MSWEP (CHIRPS) dataset. Most models show average onset dates consistent with the observational

estimates over the Sahel, except for the CNRM-CM6-1 in both protocols and the EC-Earth3P-HR in PRIMAVERA which delay the onset by about 21 and 17 days on average, respectively (Fig. 6a). This is due to a later monsoon onset over the whole Sahel region in these models, while the IPSL-CM6A-LR, EC-Earth3, ECMWF-IFS-HR and ECMWF-IFS-LR models show a dipole bias with a too late onset to the south of the box and a too early one to the north, which cancels out when averaging (not shown). Regarding the average cessation date over the Sahel, the ECMWF-IFS–HR, ECMWF-IFS-LR and especially the

EC-Earth3 advance it by about 21, 28 and 39 days, respectively (Fig. 6a). For each model, the biases for the cessation date are quite consistent over the whole Sahel region and there is no cancellation effect as for the onset in some models (not shown). This leads to very similar biases in the cessation date when averaged over the westernmost Sahel (Fig. 6b). This tendency of some models to delay the onset and of others to advance the cessation dates over the Sahel results in a majority of them underestimating the total length of the monsoon season by between 15 days (CNRM-CM6-1) and 40 days (EC-Earth3), quite

consistently across the region (not shown), the sole exception being the IPSL-CM6A-LR with a slight overestimation of about 8 days.

**3.2  Impacts of AMV on West African seasonal rainfall**

Models agree on a general enhancement of rainfall in the AMV+ simulation with respect to the AMV- one over West Africa (Fig. 7), consistent with previous studies using a similar set of simulations (Hodson et al., 2022). All models show the biggest

enhancements over the west coast, coastal parts of southern Senegal, Guinea Bisau and Guinea. Inland, the pattern of change tends to overlay the climatological values, with anomalies smaller than 0.1 mm/day to the north of the climatological contour of 1 mm/day and anomalies growing stronger to the south of this contour line. Yet, when looking at the percentage of change in rainfall relative to the climatological amount, this tendency is inverted: from changes that represent approximately 10% close to the climatological contour line of 1mm/day decreasing to the south of this line (not shown). The core of maximum positive

anomalies in JAS seasonal rainfall tends to be located at around 10ºN, close to the climatological contour values between 4 and 8 mm/day, from where they tend to decrease towards the south. They even turn negative over the West African south coast, next to the Gulf of Guinea, in the EC-Earth3P-HR and, to some extent, the IPSL-CM6-LR models. This behaviour is connected to a general response of the models to AMV by shifting northward the Atlantic ITCZ, leading to positive anomalies





over West Africa and negative ones over the Gulf of Guinea, most prominent over the ocean (not shown). Models also show
discrepancies, such as the response over the Guinea Highlands, with the CNRM-CM6-1 and, most notably, the EC-Earth3P-HR
models showing negative changes, while the rest of the models simulate an enhancement.

Taking into account the different model climatologies, the general pattern of rainfall anomalies associated with the AMV
is consistent with the one obtained in observations (compare Fig. 7 and Fig. 1c). However, the magnitudes of changes over
the Sahel tends to be underestimated in models (note that levels of the colorbar in Fig. 7 are half the ones in Fig. 1c). While
the average Sahel rainfall change between the AMV+ minus AMV- estimated from the observations is 0.35 mm/day, the
models range between 0.04 mm/day for the ECMWF-IFS-LR (only 12 % of the observation estimate) to 0.17 mm/day for the
EC-Earth3P-HR (48% of the observation estimate) (Fig. 3c,d). The underestimation is also present when comparing changes
expressed as a percentage of the climatological values (not shown).

The spread in the response in seasonal JAS rainfall of the models to the AMV pattern seems related to the spread in the
simulated climatological values, as models with weak values of the latter tend to provide weak values of the former (Fig. 3c).
The link is even stronger with the climatological number of rainy days (Fig. 3d), while it is very low with biases in the mean
intensity (correlation coefficient of 0.40). However, extrapolating the linear regression, even if models were able to provide the
highest possible observed climatological number of rainy days or total seasonal rainfall, the expected response to AMV would
still be underestimating strongly the observed estimated response (Fig. 3cd). In addition, the AMV pattern of SST depicted
in Fig. 1a also shows loads in other regions away from the North Atlantic. One such region is the Mediterranean Sea, where
warm anomalies have been shown to promote enhanced rainfall over the Sahel (e.g. Rowell, 2003; Fontaine et al., 2010, 2011;
Gaetani et al., 2010). This fact could further separate the observed estimated response from the simulated one.

### 3.3 Impacts of AMV on the intensity and number of rainy days over West Africa

All models simulate a general increase in the number of rainy days over West Africa in response to a positive AMV (Fig. 8).
Over the Gulf of Guinea changes tend to be weak or they even become negative in some models, while the increase in the
number of rainy days is stronger for the northern fringe of the JAS seasonal rainfall anomalies over the Sahel. The pattern is
similar to the estimate obtained from the observations (Fig. 1d), itself similar to the one obtained for the mean seasonal rainfall
in the same observations (Fig. 1c), consistently with Biasutti (2019). Averaged over the Sahel, model responses to AMV range
from 0.5 days in the ECMWF-IFS-LR model to 1.2 days for the CNRM-CM6-1 model in the PRIMAVERA protocol, while
the observational estimate is 1.5 days. Though models tend to underestimate the observations (ranging from 31% to 85% of the
CRU observational estimate), the underestimation is smaller than for the mean rainfall. Note that a direct comparison between
the models and the observation is hindered by the different definitions of rainy days used (the threshold for the former is 1
mm/day, while for observations is 0.1 mm/day). The patterns obtained are, however, very similar to the ones presented in Fig.
8 when using the same threshold in the simulations as in the CRU dataset to define a rainy day (0.1 mm/day) (not shown), with
a range of averaged impact over the Sahel between 0.5 days in the ECMWF-IFS-HR model to 1.5 days for the CNRM-CM6-1
model in the PRIMAVERA protocol.



The models' responses to a positive AMV also include an increase of the mean intensity of rainfall over West Africa (Fig. 9). Unlike the impact on the number of rainy days, the maximum changes in the intensity tend to be simulated more to the south, where the mean JAS seasonal rainfall anomalies are strongest. Averaged over the Sahel, the change in intensity shows a strong spread, with a multi-model mean of 0.11 mm/day but a range from the very weak changes of 0.02 mm/day to 0.19 mm/day simulated by the ECMWF-IFS-LR and the CNRM-CM6-1 in the DCPP-C protocol, respectively.

According to the decomposition proposed in equation 3, the changes in the number of rainy days tend to dominate the changes in the total seasonal rainfall over the northern fringe, where the changes in the number of rainy days tend to be stronger and those in seasonal rainfall weaker (see hatching in Fig. 8). Conversely, changes in total rainfall seasonal amounts tend to be dominated by the changes in the mean intensity of rainfall more to the south, typically south of 15ºN or 10ºN, depending on the model, where the maximum anomalies of total seasonal rainfall are located (see hatching in Fig. 9). In Fig. 10 we show this behavior more clearly: for each grid point in the West African region (see blue box in Fig. 1cd), we calculate the percentage of explained JAS seasonal rainfall anomaly by the change in the number of rainy days as the ratio of the first term on the right hand to the left hand side of equation 3. We then divide the range of JAS seasonal rainfall anomalies in 6 intervals and for each model, we gather together all grid points that belong to a given interval and plot the median of the explained percentage (Fig. 10a). The same procedure is followed for the change in the mean intensity, which is shown in Fig. 10b. There is a clear tendency in all models for the change in the number of days to dominate over regions with weak changes of JAS seasonal rainfall (Fig. 10a) and for the change in the mean intensity to dominate in the regions of high values of JAS seasonal rainfall anomalies (Fig. 10b). To evaluate the accuracy of the decomposition, for each grid point we calculate the total percentage of explained JAS seasonal rainfall anomalies by and as the ratio of the right hand side to the left hand side of equation 3. We divide this percentage in different intervals and show in Fig. 10c for each model the percentage of area over West Africa covered by each interval where anomalies of seasonal rainfall were statistically significant (Fig. 7). This percentage of area is calculated as the ratio of grid points in which the percentage of explained variance belongs to a given interval to the total number of grid points with statistically significant seasonal rainfall anomalies over West Africa (see blue box in Fig. 1cd). The decomposition proposed in equation 3 is a good approximation for most models: over most of West African grid points where seasonal rainfall changes are statistically significant, the right hand side of equation 3 explains close to 100 % of its left hand side. The main exceptions are the CNRM-CM6-1 model in the DCPP-C protocol and the EC-Earth3P-HR, where the approximation is over-representing and under-representing, respectively, the change in total JAS seasonal rainfall.

### 3.4 Impacts of AMV on the frequency of moderate, heavy and extreme precipitation events

In response to AMV, models show an enhancement in the number of moderate rainfall events (i.e. those below the 75 percentile taking into account only rainy days) over the northern part of the Sahel, lying over regions of weak JAS seasonal rainfall anomalies, reaching values above 1 day per season at some grid points (Fig. 11). The average change in the Sahel box is, however, smaller, as models tend to simulate negative anomalies (i.e. less moderate events) to the south, some clearly affecting the southern part of the Sahel box (as the IPSL-CM6A-LR, CNRM-CM6-1 in both protocols and EC-Earth3P-HR). Conversely, the number of heavy rainfall events (those above the 75 percentile) shows a general enhancement over West Africa with the





maximum values aligned over the ones of JAS seasonal rainfall (Fig. 12). On average, over the Sahel box, the increase in the number of heavy events ranges between 0.17 and 0.92 days per season for the ECMWF-IFS-LR and IPSL-CM6A-LR models, respectively.

Note that, by definition, the changes in the number of moderate events (Fig. 11) plus those coming from changes in heavy
events (Fig. 12) are equal to the changes in the total number of rainy days (Fig. 8). Towards the southern limit of the Sahel, where the main JAS seasonal rainfall changes are simulated, the enhancement in the frequency of heavy events is higher than the one for moderate events, which can even be reduced in some models south of 10ºN (Fig. 11). The dominance of the changes in the number of heavy events in regions where seasonal rainfall changes are strong is clearly seen in Fig. 13b, where, similarly to what was presented in Fig. 10, results are shown as a function of the interval of JAS seasonal rainfall change. The last
interval includes events up to 0.55 mm/day, because most grid points over West Africa show JAS seasonal rainfall anomalies weaker than this value (Fig. 13c). The tendency of models to reduce the number of moderate events over those regions is also highlighted in Fig. 13a. The comparison of both plots suggests that over grid points where strong anomalies of seasonal rainfall are simulated (typically over the southern Sahel limit towards 10ºN), models are providing changes in the distribution of rainfall events, without strong changes of the total number of rainy days (Fig. 8). This provides an explanation as to why in
those areas the changes in the mean intensity tend to dominate those coming from the change in the number of rainy days in the overall response of JAS seasonal rainfall (Fig. 9 and Fig. 10ab): in response to a positive phase of AMV the models tend to simulate a shift in the rainfall distribution towards higher values over the southern margins of the Sahel, providing higher rainfall amounts when it rains.

Conversely, over regions where the response to AMV in the JAS seasonal rainfall is small (the northern part of the Sahel),
both moderate and heavy events are enhanced (Fig. 13ab), with a slight dominance of the former (compare the values lying between contour lines of 0.05 mm/day and 0.10 mm/day in Figs. 11 and 12). This suggests that over those regions the increase in JAS seasonal rainfall comes from an increase of all types of rainy days, without strong changes in the distribution of rainfall events. Thus, in those regions the changes in JAS seasonal rainfall are dominated by the general increased occurrence frequency of rainy days (Fig. 8 and Fig. 10a).

Regarding the occurrence of extreme rainfall events (those defined using the 95 percentile threshold for all rainy days), in response to a positive phase of the AMV models also simulate a general enhancement over West Africa, affecting the southern part of the Sahel (Fig. 14). On average over the Sahel box, changes range from 0.03 to 0.27 days per season in the ECMWF-IFS-LR and IPSL-CM6-LR models, respectively, with a multi-model mean change of 0.15 days per season.

### 3.5   Impacts of AMV on the timing of the monsoon season

The models show no clear impact of AMV on the monsoon onset over the whole West Africa, with mostly non statistically significant signals of different signs (not shown). Over the Sahel box, only the ECMWF-IFS-LR model simulates a statistically significant earlier onset of 1.7 days (Fig. 6a). There is more consistency regarding the impact of the positive phase of AMV with respect to the negative one on the monsoon demise date over West Africa, with most models simulating a later cessation date over most of West Africa, albeit with different magnitudes and degrees of statistical significance (not shown). Over the



Sahel box, all models simulate a later demise date for the AMV+ experiment compared to the AMV- one, which ranges from
1.3 to 5.0 days for the ECMWF-IFS-LR and EC-Earth3P-HR models, respectively. However, the differences are statistically
significant for only half the models (Fig. 6a). There is also a consistent increase in the length of the season as a response to a
positive AMV over most of West Africa (not shown). In the Sahel box, this increase in the monsoon season ranges from 1.5 to
5.3 days in the EC-Earth3 and ECMWF-IFS-HR models, respectively, and is statistically significant for most models (Fig. 6a).
The impact of AMV on the seasonality of the monsoon is stronger and more consistent over the Sahel latitudes west of 10ºW,
with all models simulating a statistically significant earlier onset, later cessation and longer monsoon season in the AMV+
experiment compared to the AMV- one (Fig. 6b).

## 4   Summary and discussion

Previous studies agree that in response to a positive phase of AMV total seasonal rainfall amounts over the Sahel tend to
increase (e.g. Folland et al., 1986; Knight et al., 2006; Zhang and Delworth, 2006; Mohino et al., 2011; Ting et al., 2011;
Martin and Thorncroft, 2014; Martin et al., 2014; Villamayor et al., 2018b; Hodson et al., 2022). Here we go beyond the total
seasonal amounts and investigate the influence of AMV on the intraseasonal characteristics of rainfall by analyzing a set of
model simulations under a largely common protocol. The experiments consist of 10 year-long runs with SST in the North
Atlantic forced to resemble an idealized AMV pattern. We analyze the biases shown by the models and estimate the impact
of AMV by comparing the 10-year averaged AMV+ and AMV- experiments. The protocol followed allows us to focus on the
impact of the AMV pattern while other sources of long-term variability are removed. In addition, the 10-year averages are
consistent with the decadal time scale of AMV which shows decadal-long periods with values above 1 standard deviation of
the index (Fig. 1b).

  Models show consistent bias patterns in the JAS seasonal total rainfall amounts, number of rainy days and mean rainfall
intensity, with an underestimation over the Sahel and an overestimation to the south, especially over the Guinea Coast. Over
West Africa, the biases are well above the observational uncertainty in the case of the total rainfall values. However, the rainfall
mean intensity clearly exceeds the observational uncertainty only over the Sahel, with a clear underestimation. For the number
of rainy days, only the positive biases over the Gulf of Guinea are well above the observational range for all the models, while
the underestimation over the Sahel is only clear for roughly half the models. Most models underestimate the average length of
the rainy season over the Sahel, some due to a too late monsoon onset and others due to a too early cessation. We note these
biases have been calculated as the subtraction of the current climatological observed values to those obtained by averaging the
AMV+ and AMV- experiments, which, in the case of the DCPP-C protocol, provides a climatological background consistent
with pre-industrial conditions. Therefore, the biases estimated for the DCPP-C protocol could have an added effect due to the
different climatological backgrounds being considered. However, the comparison of the biases estimated for the CNRM-CM6-
1 model for both protocols suggest that this effect is negligible for the rainfall estimates shown in this work.

  Despite differences in the amplitudes of changes, the models analyzed show high agreement in the response of West African
rainfall to a positive phase of the AMV. This response involves a general increase in JAS seasonal rainfall amounts that overlays





the climatological values, with higher changes in the southern Sahel, typically close to 10ºN, and weaker ones to the North.
The latter are mainly related to an increase in the number of rainy days due to the enhancement of all types of rainfall events,
moderate, heavy and extreme. The stronger changes observed in the southern part of the Sahel are better explained by an
increase in the mean intensity of rainfall, as the number of heavy and extreme rainfall events grows, while those for moderate
changes little or it even decreases. This enhancement in the amount and intensity of rainfall in response to the positive phase
of AMV can have potentially societal impacts, as an increase in crop yield in semi-arid Sahel (Guan et al., 2015), but also in
flood risk (Tazen et al., 2019; Elagib et al., 2021).

Models show less impact of AMV on the timing of the monsoon over the Sahel box (Fig. 1a). Most of them suggest a
lengthening of the monsoon season in the AMV+ experiment compared to the AMV- one, principally due to a later demise. As
much of the rainfall falling over the Sahel comes from local recycling (Nieto et al., 2006), this higher consistency in the demise
date could be related to increased soil moisture following an enhanced rainfall season in response to a positive AMV phase.
The more humid soil could, through land-atmosphere interactions, provide the source of moisture to increase precipitation at
the end of the season and thus allow a later demise date over the Sahel. Conversely, models show high consistency on the
response of the westernmost Sahel to the positive phase of the AMV, with an earlier onset, later cessation and longer monsoon
length (Fig. 1b).

Regarding the differences in the amplitude of changes over the Sahel box, two clusters of models can be described. One
includes the ECMWF-IFS-HR, EC-Earth3 and, especially, the ECMWF-IFS-LR models. This cluster shows weak changes in
most of the metrics analyzed: the total JAS seasonal anomalies, mean intensity, number of rainy days, and on the frequency of
heavy and extreme rainfall events. On the other hand, the IPSL-CM6A-LR, EC-Earth3P-HR and the CNRM-CM6-1 under both
protocols tend to show larger changes in the above metrics. This second cluster is also the one that shows a longer monsoon
season due to a later cessation date (Fig. 6a). These two clusters can also be distinguished in their climatological values of total
JAS seasonal rainfall and total number of rainy days, with weaker values for the first group and stronger values for the second
one (Fig. 3b), suggesting biases could affect the simulated responses. However, we note that the number of models analyzed
is small, so it is difficult to extract robust conclusions in this regard. In addition, the sample of models used in this work has
little diversity: all use the Nucleus for European Modelling of the Ocean (NEMO) for their ocean component, with all sharing
version 3.6 (Madec et al., 2017) except the ECMWF-IFS-HR and ECMWF-IF-LR, which use version 3.4. And for most of
them, the atmospheric component is related to the Integrated Forecasting System (IFS).

The above results and grouping also suggest that the differences in the protocols followed, DCPP-C versus PRIMAVERA,
are of second order importance to the impact simulated by the models. Once the amplitude of the forcing AMV pattern is
accounted for, the resemblance of the AMV impact simulated by the CNRM-CM6-1 model under both protocols is higher than
the one provided by any other model following the same protocol, in agreement with the analysis of a similar set of simulations
(Ruprich-Robert et al., 2021). In addition, there is no clear relation between the horizontal resolution of the atmospheric com-
ponent and the simulated impact of AMV on West African rainfall characteristics. The previous grouping does not distinguish
between higher and lower resolution models in the range span by our analysis (approximately 0.25º to 2.5º). In fact, in the
first group we find two versions of the same model run at different resolutions, and the second cluster groups together the



model with the highest and lowest resolutions, the EC-Earth3P-HR and the IPSL-CM6-LR models, respectively. This result is consistent with the generally small effect of model resolution on the simulated AMV impacts shown by Hodson et al. (2022),

particularly on rainfall over West Africa. However, the highest horizontal resolution in the models analyzed was still below the convection-permitting capabilities of other models for which a more realistic climatology and stronger impact on extremes has been reported (Berthou et al., 2019; Kendon et al., 2019).

Our results are consistent with previous studies regarding the enhancement of seasonal summer rainfall over the Sahel in response to AMV (e.g. Folland et al., 1986; Knight et al., 2006; Zhang and Delworth, 2006; Mohino et al., 2011; Ting et al.,

2011; Martin and Thorncroft, 2014; Martin et al., 2014; Villamayor et al., 2018b; Hodson et al., 2022). There is, however, a notable exception to the general agreement between the observed and simulated pattern of impact of AMV on West African seasonal JAS rainfall shown in this study. Over the westernmost coast models simulate a strong enhancement of rainfall in response to AMV, while values are weak and not even statistically significant in the observations estimated from CRU (compare Fig. 7 and Fig. 1c). This could suggest a systematic bias common to all analysed models. However, it could also be related to

the statistically significant positive trend in the AMV index used for the observational estimate presented in Fig. 1c. Long-term trend variations of Sahel rainfall have shown a decoupling between the west, more prone to drought, and the central and east regions, more prone to an enhancement both during the instrumental period and in climate projections (Lebel and Ali, 2009; Mohino et al., 2011; Monerie et al., 2020a, b). When other AMV indices that do not have a long-term trend are used, the regression patterns of observed seasonal rainfall seem more similar to the ones provided by the simulations, with the strongest

loads over the westernmost coast (Mohino et al., 2011). This suggests that the disparity shown in this study would come more from the lack of an SST signal in the boundary conditions related to the long-term trend shown by the observed AMV index in Fig. 1b than from a clear failure of the models to simulate AMV impacts.

Our analysis also agrees with the observational results from Badji et al. (2022) on the positive link found between AMV and the number of rainy days, mean intensity and occurrence of heavy and extreme rainfall events over the western Sahel. However,

while all models show a statistically significant response of the monsoon timing over the western Sahel to the AMV phase (Fig 6b), Badji et al. (2022) reported no clear response. Several factors could contribute to this disparity. Firstly, model deficiencies could prevent them from properly reproducing the impact of AMV on the timing of the monsoon, though the high consistency among models suggests other factors might be at play. Secondly, there could be other sources of decadal variability, as the Interdecadal Pacific Oscillation signaled by Badji et al. (2022), which could contribute more strongly than AMV and mask the

influence of the latter. Thirdly, there is an uncertainty coming from the definition of the onset and cessation dates, which is different in both studies. In fact, Badji et al. (2022) report slightly different results when a different definition of the monsoon timing is used.

Despite the general agreement between model results and observed estimates, models clearly underestimate the amplitudes of the changes in seasonal rainfall amounts (Fig. 3ab). This is consistent with previous works showing that atmosphere gen-

eral circulation models tend to underestimate the response of West African rainfall to anomalous SSTs (e.g. Joly et al., 2007; Rodriguez-Fonseca et al., 2011; Vellinga et al., 2016; Villamayor and Mohino, 2015). The lack of representation of some processes and feedbacks could hinder the simulation of the correct amplitude of the impacts of SST changes on West African



rainfall (Yu et al., 2016; Balkanski et al., 2021). Comparison with the results from Badji et al. (2022) suggests this underestimation also affects other indices. Part of the disparity could be inherent to the comparison of station data with grid points that represent larger areas. Our results further suggest that model biases could also be contributing to this underestimation (Fig. 3cd). The underrepresentation of other sources of decadal timescale variability, like SST variability in different regions and the direct (i.e. not ocean-mediated) impact of concurrent radiative forcings (Mohino et al., 2011; Hirasawa et al., 2020), which vary in the observations but are fixed in the simulations, could further separate the observed changes from the simulated ones. Additionally, there is an uncertainty coming from the shortness of the observed record. Note also that we are comparing the observed transient response to AMV with the models' response to a persistent SST pattern, for which noise is further filtered out by ensemble averaging.

The experiments analysed are not exempted from potential problems related to SST restoring techniques. O'Reilly et al. (2023) suggest that restoring SSTs in the Tropical North Atlantic can lead to year-mean exaggerated responses through an unrealistic local release of surface heat fluxes into the atmosphere following a positive AMV phase. However, this inconsistency manifests itself primarily during boreal winter, suggesting that the boreal summer response to AMV analysed here would be more consistent with results from free (i.e. unrestored) simulations. Furthermore, as discussed above, the comparison with observations suggest that these experiments, far from exaggerating, actually underestimate the observed AMV impact, even accounting for model biases.

Our results suggest that the observed warming of the North Atlantic SSTs related to the AMV since the 1980s (Fig. 1) could have contributed to the positive trend in the occurrence of extreme events over the Sahel reported in different studies (i.e. Taylor et al., 2017). However, the precise extent of this contribution would need a more detailed analysis taking into account other possible sources of variability. Additionally, the AMV pattern used in the experiments analysed in this work shows prominent loads both in the tropical and extratropical North Atlantic. The contribution of each part to the total AMV impact on West Africa and the linearity of the addition of these contributions could be further explored under the DCCP-C protocol by analysing the tropical and extratropical AMV experiments (Boer et al., 2016).

Lastly, our results might have implications for decadal prediction. The AMV has been shown to be highly predictable at multiyear to decadal time scales (Kim et al., 2012; Doblas-Reyes et al., 2013; Delgado-Torres et al., 2022). Given that some models are able to predict changes in summer seasonal rainfall totals over the Sahel at decadal timescales and that the main mechanism to explain this potential comes from the AMV (e.g. Gaetani and Mohino, 2013; Mohino et al., 2016; Sheen et al., 2017), our study suggests a potential for decadal prediction systems to also predict changes in the intraseasonal characteristics of rainfall over the Sahel, including the occurrence of extreme events.

*Author contributions.* EM, P-AM, JM, MD and FD-R discussed the initial conception of the work; EM, P-AM, JM and MD participated in the methodological design; CDR provided guidance on the model runs; EM did the formal analysis and wrote the original draft; all authors reviewed and edited.



*Competing interests.* The authors declare that they have no known competing financial interests or personal relationships that could have appeared to influence the work reported in this paper.

*Acknowledgements.* Authors acknowledge the use of JASMIN facilities for the PRIMAVERA data as part of the IS-ENES3 project that has received funding from the European Union's Horizon 2020 research and innovation programme under grant agreement No 824084. We acknowledge the World Climate Research Programme, which, through its Working Group on Coupled Modelling, coordinated and promoted
CMIP6. We thank the climate modeling groups for producing and making available their model output, the Earth System Grid Federation (ESGF) for archiving the data and providing access, and the multiple funding agencies who support CMIP6 and ESGF. The observational data sources used in this study were: ERSSTv4 obtained through https://www1.ncdc.noaa.gov/pub/data/cmb/ersst/v4/netcdf/ (last accessed 30 nov 2022), MSWEP v2.8 obtained through http://www.gloh2o.org/mswep/ (last accessed 29 nov 2022) and CHIRPS v2.0 obtained through https://data.chc.ucsb.edu/products/CHIRPS-2.0/ (last accessed 29 nov 2022). We also acknowledge the Climatic Research Unit (University of
East Anglia) and Met Office for providing CRU TS version 4.06, which was obtained through https://crudata.uea.ac.uk/cru/data/hrg/cru_ts_4.06/ (last accessed 29 nov 2022).

Funding: EM acknowledges the funding provided by the Spanish Ministry of Science and Innovation DISTROPIA project [grant number PID2021-125806NB-I00]. JM acknowledges the support of the JPI climate/JPI ocean project ROADMAP [grand number ANR-19-JPOC-003] and the ARCHANGE project of the "Make our planet great again" program [grant number ANR-18-MPGA-0001, France].



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

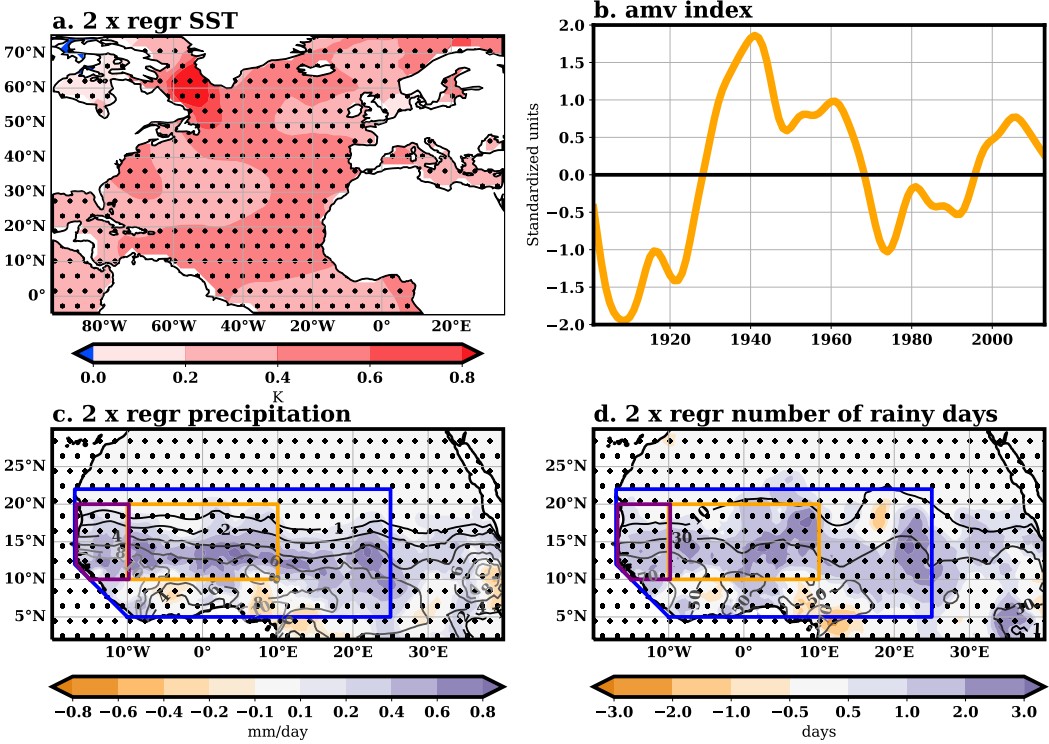

**Figure 1.** AMV and impacts in observations in the 1901-2013 period: Twice the regression onto the standardized AMV index of ERSSTv4 year-averaged SSTs onto the AMV index (a, shaded in K); standardized AMV index (as shown in Boer et al., 2016) (b); the mean seasonal July to September rainfall (c, shaded in mm/day) and the total number of rainy days in the season (d, shaded in days). In plots c-d regression is only shown over regions where the correlation between the AMV index and the field are statistically significant (p<0.05). In plots c and d climatological values are shown in contours. The orange, purple and blue boxes in plots c and d mark the Sahel, westernmost Sahel and West African region as defined in this study.





**Figure 2.** Biases in mean seasonal July to September (JAS) rainfall simulated by the models (mm/day, shaded) with respect to MSWEP estimate in the period 1979-2020. As an estimate of observation uncertainty, plot h shows the difference between CHIRPS (1981-2021) and MSWEP seasonal rainfall. In all plots, contours mark the MSWEP JAS seasonal rainfall (contour values of 1, 2, 4, 6, 8 and 10 mm/day) interpolated into the model's grid (or CHIRPS). The simulations following the PRIMAVERA protocol are marked as blue in the model name labels.



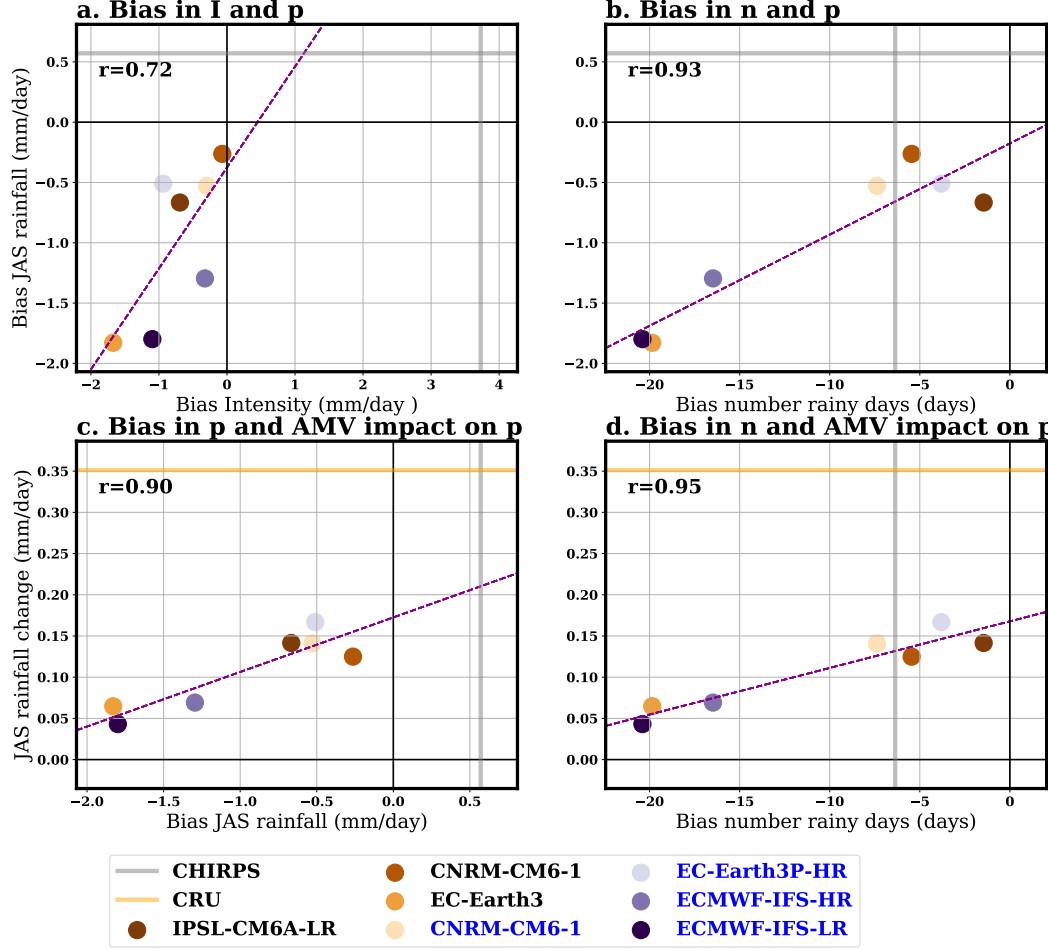

**Figure 3.** Scatterplot of box averages over the Sahel (defined as the region between 10ºW-10ºE, 10ºN-20ºN, see red box in Fig. 1cd): first line: biases in JAS mean rainfall (mm/day) versus biases in mean intensity in rainy days (mm/day) (a) and the number of rainy days (days) (b). Second line: AMV+ minus AMV- response in JAS mean rainfall (mm/day) versus biases in JAS mean rainfall (mm/day) (c) and the number of rainy days (days) (d). Biases are calculated as differences to the MSWEP dataset. For reference, the average mean JAS rainfall, mean intensity of rainfall and number of rainy days in this dataset in the 1970-2020 period is 3.0 mm/day, 6.1 mm/day and 37 days, respectively. For simulations following the PRIMAVERA protocol (marked as blue in the model name labels) only half the anomalous values are shown for the AMV response. The dashed line in each plot shows the least-squares linear fit. The corresponding correlation coefficient is shown at the top left of each plot. Observational estimates for CHIRPS and CRU are shown in the plots as gray and orange lines, respectively.

**Figure 4.** Biases in the number of rainy days per JAS season simulated by the models (day, shaded) with respect to MSWEP estimates in the period 1979-2020. As an estimate of observational uncertainty, plot h shows the difference between CHIRPS (1981-2021) and MSWEP seasonal number of rainy days. In all plots, contours mark the MSWEP JAS number of days per season (contour values every 20 days starting from 10 days). The simulations following the PRIMAVERA protocol are marked as blue in the model name labels.





**Figure 5.** Biases in the mean intensity of rainfall (taking into account only rainy days) in JAS simulated by the models (day, shaded) with respect to MSWEP estimates in the period 1979-2020. As an estimate of observational uncertainty, plot h shows the difference between CHIRPS (1981-2021) and MSWEP mean intensity of rainfall. In all plots, contours mark the MSWEP JAS number of days per season (contour values every 2 mm/day starting from 4 up to 12 mm/day). The simulations following the PRIMAVERA protocol are marked as blue in the model name labels.



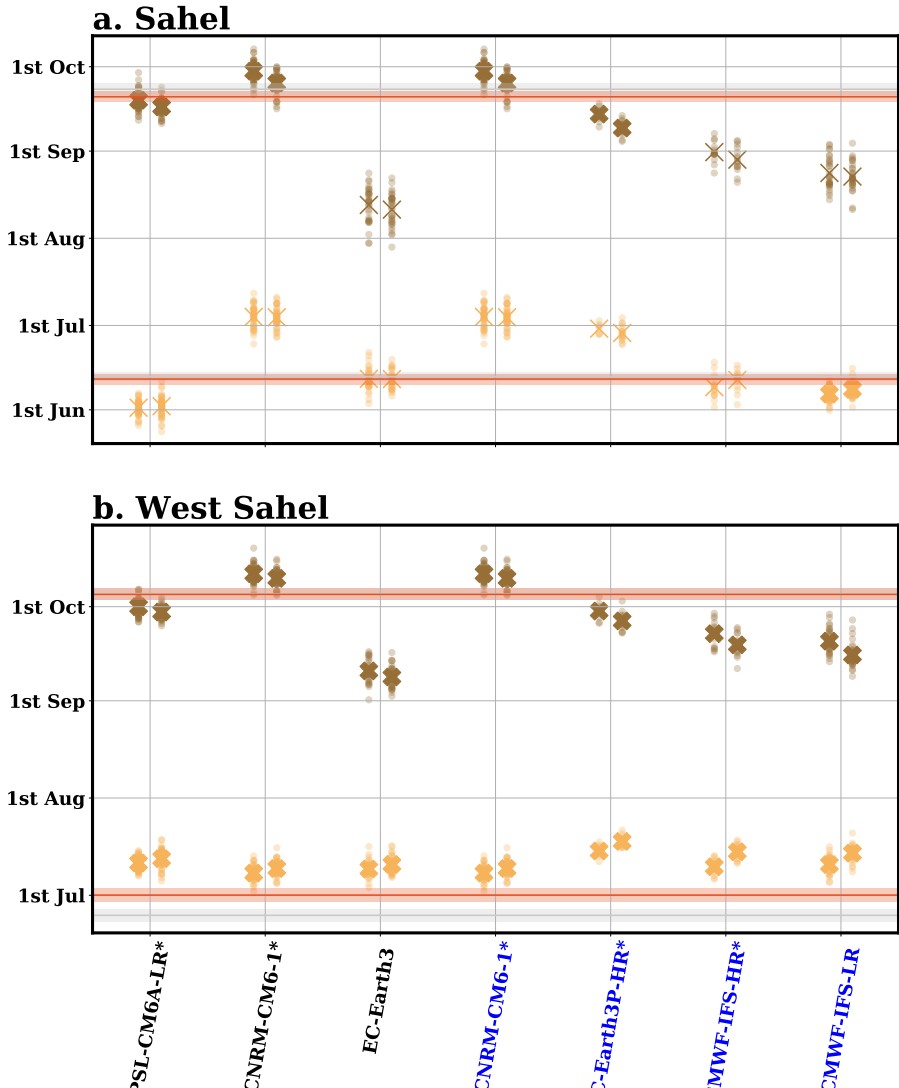

**Figure 6.** Dates for the monsoon onset (orange) and demise (brown) in models averaged over the a) Sahel (defined as the area 10ºW-10ºE, 10ºN-20ºN, see red box in Fig. 1cd) and b) Westernmost Sahel (17ºW-10ºE, 10ºN-20ºN, after removal of the area west of the line connecting the points 17ºW-12ºN and 15ºW-10ºN, see green box in Fig. 1cd) for the simulations. PRIMAVERA simulations are marked as blue in the model labels. The values for the AMV+ simulation are shown slightly to the left of the corresponding axis for each model, while the ones for the AMV- simulation are located to the right. Small circles show the averages for the 10 years in each ensemble member, while big crosses show the values averaged over all ensemble members. Crosses are bold if the AMV+ minus AMV- differences are statistically significant (p < 0.05). Values of monsoon onset and demise are shown for the rainfall estimates obtained from MSWEPv280 (red line, climatology 1979-2020) and CHIRPS (gray line, climatology 1981-2021). Shading indicates the standard deviation of 10 year running mean values (so that they can be compared with 10 year means for each ensemble). The asterisk symbol in the names of the models indicate that the AMV+ minus AMV- difference in the length of the monsoon are statistically significant (p<0.05) averaged over the Sahel box. The differences in the length of the season for the westernmost Sahel box are all statistically significant (p<0.05).





**Figure 7.** Difference in mean JAS seasonal rainfall between AMV+ and AMV- experiments (shaded, mm/day). For simulations under PRIMAVERA protocol (marked as blue in the model name labels) only half the anomalous values are shown. Contours mark the climatological values of mean JAS seasonal rainfall (contour values of 1, 2, 4, 6, 8 and 10 mm/day). Regions where differences are not statistically significant (p<0.05) are dotted.





**Figure 8.** Difference in the number of rainy days between AMV+ and AMV- experiments (shaded, days). For simulations under PRIMAV-ERA protocol (marked as blue in the model name labels) only half the anomalous values are shown. Contours mark the corresponding differences in mean JAS seasonal rainfall (contour values of 0.05, 0.1, 0.2 and 0.4 in mm/day, solid for positive values and dashed for negative ones). Regions where differences in the number of rainy days are not statistically significant (p<0.05) are dotted. Hatching by straight lines mark regions where the difference in the number of rainy days explains most of the difference in mean JAS seasonal rainfall (i.e. where the first term in the right hand side of equation 3 dominates).



**Figure 9.** Difference in the mean rainfall intensity between AMV+ and AMV- experiments (shaded, days). For simulations under PRI-MAVERA protocol (marked as blue in the model name labels) only half the anomalous values are shown. Contours mark the corresponding differences in mean JAS seasonal rainfall (contour values of 0.05, 0.1, 0.2 and 0.4 in mm/day, solid for positive values and dashed for negative ones). Regions where differences in the mean rainfall intensity are not statistically significant (p<0.05) are dotted. Hatching by straight lines mark regions where the difference in the mean rainfall intensity explains most of the difference in mean JAS seasonal rainfall (i.e. where the second term in the right hand side of equation 3 dominates).

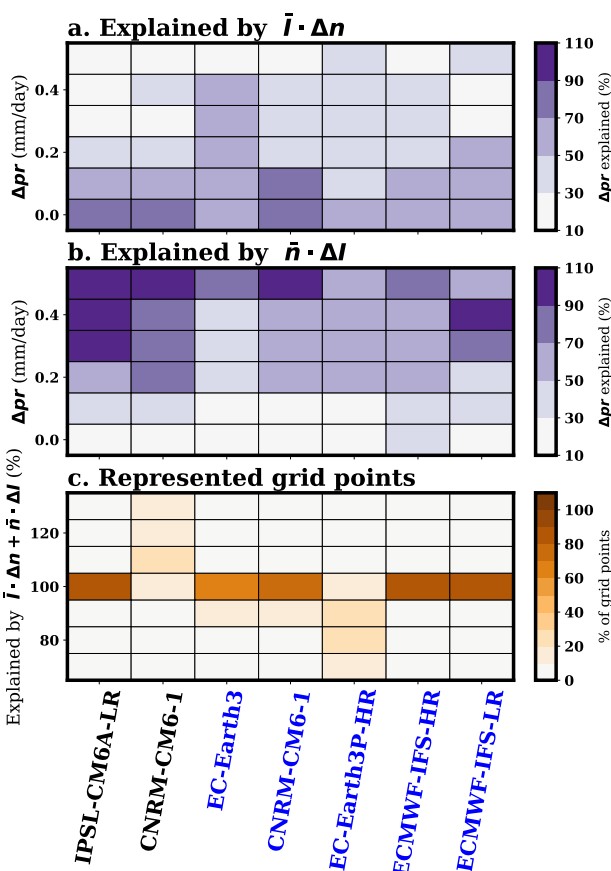

**Figure 10.** a) Percentage of explained difference in mean JAS seasonal rainfall between AMV+ and AMV- experiments in West Africa (see region marked in blue in Fig. 1) by changes in the number of rainy days (first term in the right hand side of equation 3) as a function of the differences in rainfall for each model. For each model all grid points that belong to the rainfall difference interval (Y axis) are taken into account and the median value is shown (color, %) b) Same as a) but explained by changes in the mean intensity (second term in the right hand side of equation 3). c) Area of West Africa where seasonal rainfall changes are statistically significant (in % of grid points) per model covered as function of the differences in mean JAS seasonal rainfall explained together by the changes in the number of days and in the intensity of rainfall (i.e. ratio of the right hand part of equation 3 to the left hand one). For simulations under PRIMAVERA protocol (marked as blue in the model name labels) only half the anomalous values are taken into account for the calculations.



**Figure 11.** Difference in the number of moderate rainy days (i.e. those below the 75 percentile of rainy days) between AMV+ and AMV-experiments (shaded, days). For simulations under PRIMAVERA protocol (marked as blue in the model name labels) only half the anomalous values are shown. Contours mark the corresponding differences in mean JAS seasonal rainfall (contour values of 0.05, 0.1, 0.2 and 0.4 in mm/day, solid for positive values and dashed for negative ones). Regions where differences in the number of rainy days are not statistically significant (p<0.05) are dotted.



**Figure 12.** Difference in the number of heavy rainy days (i.e. those above the 75 percentile of rainy days) between AMV+ and AMV-experiments (shaded, days). For simulations under PRIMAVERA protocol (marked as blue in the model name labels) only half the anomalous values are shown. Contours mark the corresponding differences in mean JAS seasonal rainfall (contour values of 0.05, 0.1, 0.2 and 0.4 in mm/day, solid for positive values and dashed for negative ones). Regions where differences in the number of rainy days are not statistically significant (p<0.05) are dotted.

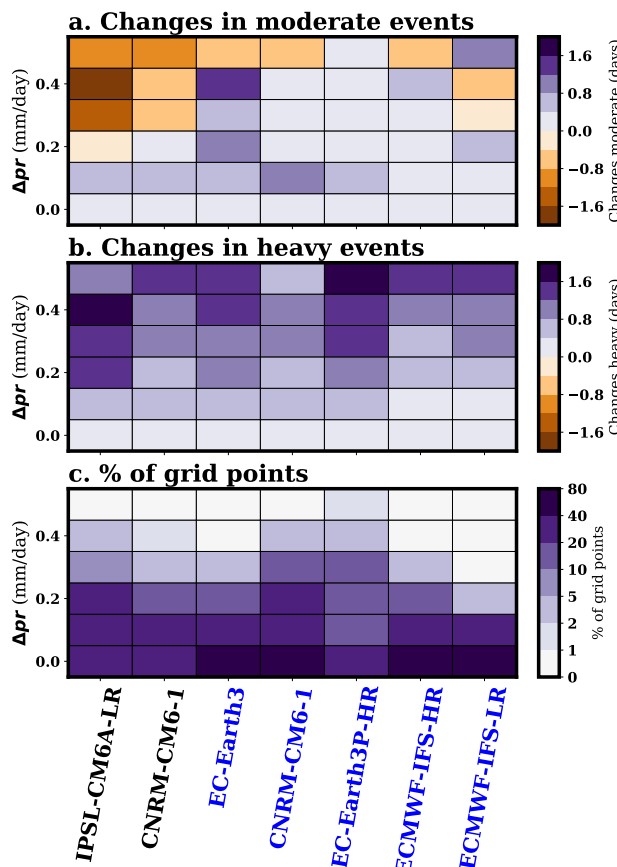

**Figure 13.** a) Differences in the number of moderate rainy days (i.e. those below the 75 percentile) between AMV+ and AMV- experiments in West Africa (see region marked in blue in Fig. 1) as a function of the differences in JAS seasonal rainfall for each model. For each model all grid points that belong to the rainfall difference interval (Y axis) are taken into account and the median value is shown (color, days). b) Same as a) but for the number of heavy rainy days (i.e. those above the 75 percentile). c) Percentage of grid points per model covered as function of the differences in mean JAS seasonal rainfall. For simulations under PRIMAVERA protocol (marked as blue in the model name labels) only half the anomalous values are taken into account for the calculations.





**Figure 14.** Difference in the number of extreme rainy days (i.e. those above the 95 percentile of rainy days) between AMV+ and AMV-experiments (shaded, days). For simulations under PRIMAVERA protocol (marked as blue in the model name labels) only half the anomalous values are shown. Contours mark the corresponding differences in mean JAS seasonal rainfall (contour values of 0.05, 0.1, 0.2 and 0.4 in mm/day, solid for positive values and dashed for negative ones). Regions where differences in the number of rainy days are not statistically significant (p<0.05) are dotted.



**Table 1.** List of models analyzed, along with their atmospheric horizontal resolution (longitude x latitude), protocol followed, ensemble size for the positive/negative experiment, main reference for the model documentation and Institution involved.

| Model | Resolution | Protocol | Members | Reference | Modelling group |
|---|---|---|---|---|---|
| IPSL-CM6-LR | 2.5º x 1.3º | DCPP-C | 50 / 50 | Boucher et al. (2020) | IPSL |
| CNRM-CM6-1 | 1.4º x 1.4º | DCPP-C | 40 / 40 | Voldoire et al. (2019) | CNRM, CERFACS |
| EC-Earth3 | 0.7º x 0.7º | DCPP-C | 32 / 32 | Döscher et al. (2022) | EC-Earth-Consortium |
| CNRM-CM6-1 | 1.4º x 1.4º | PRIMAVERA | 15 / 15 | Voldoire et al. (2019) | CNRM, CERFACS |
| EC-Earth3P-HR | 0.35º x 0.35º | PRIMAVERA | 7/17 | Haarsma et al. (2020) | EC-Earth-Consortium |
| ECMWF-IFS-HR | 25 km | PRIMAVERA | 15 / 15 | Roberts et al. (2018) | ECMWF |
| ECMWF-IFS-LR | 50 km | PRIMAVERA | 30 / 30 | Roberts et al. (2018) | ECMWF |