# Peer review of "Impact of AMV on rainfall intensity distribution and timing of the West African Monsoon in DCPP-C-like simulations"

_EGUsphere, 2023_

## Author Comment (AC1)

Review of the manuscript 'Impact of AMV on rainfall intensity distribution and timing of the West African Monsoon in DCPP-C-like simulations' by Mohino et al.

The authors examine the influence of the Atlantic Multidecadal Variability (AMV) on intraseasonal precipitation characteristics by analyzing a series of model simulations following a commonly used protocol. They analyze the biases shown by the models and estimate the impact of AMV by comparing 10-year averaged AMV+ and AMV- experiments. Models show consistent bias patterns in the summer JAS seasonal total rainfall amounts, number of rainy days and mean rainfall intensity, with an underestimation over the Sahel and an overestimation to the south, especially over the Guinea Coast. The models analyzed show high agreement in the response of West African rainfall to a positive phase of the AMV. This response involves a general increase in JAS seasonal rainfall amounts with higher changes in the southern Sahel, typically close to 10N, and weaker ones to the North. The latter are mainly related to an increase in the number of rainy days due to the enhancement of all types of rainfall events, moderate, heavy and extreme. The stronger changes observed in the southern part of the Sahel are better explained by an increase in the mean intensity of rainfall, as the number of heavy and extreme rainfall events grows, while those for moderate changes little or it even decreases.

We thank the reviewer for her/his careful reading of the manuscript and for the remarks and suggestions. In the following we provide discussion on these comments using red to distinguish our answers from the reviewer's comments.

Comments:

- introduction: causes of the AMV: please note that AMV variability might also be caused in a model that does not include ocean circulation changes. Please see Clement et al. DOI: 10.1126/science.aab398

Thanks for this remark. We have added the following sentence on this possible mechanism for producing AMV as the second sentence in the second paragraph of the introduction (starting in line 30): "Another possible mechanism, also internal to the climate system, is the response of the upper ocean mixed layer to mid-latitude atmospheric stochastic forcing and subsequent thermal coupling in the tropics (Clement et al. 2015)."

- method: onset and demise of the wet season: 'The daily rainfall anomaly is obtained as the rainfall for each day minus the long-term climatological mean daily rainfall using all available years in the observations and all years and ensemble members in both experiments, AMV+ and AMV-, in the models.' I find this description confusing. I would prefer to rewrite the sentence. Do you mean: The daily rainfall anomaly is obtained as the climatological rainfall for each day minus the annual mean rainfall using ..…

Sorry that the description was confusing. As in Liebmann et al. (2012), onset and demise dates are calculated separately for each year (of each ensemble member, in the case of the simulations), using the actual rainfall value for each day of that year (and ensemble member, in the case of the simulations). The daily rainfall anomaly is then calculated for each day of each year (and ensemble member, in the case of the simulations) as the difference between the actual daily rainfall and the long-term annual mean rainfall, which is defined as the averaged daily rainfall using all days and all years (and ensemble members in the case of the simulations). We have modified two sentences in the paragraph to highlight this detail of the calculation being performed for each year. The sentences in lines 171-174 now read as: "It consists in calculating for each calendar year the dates for the

minimum and maximum of the daily cumulative rainfall anomaly which provide the onset and cessation dates, respectively, for the season of that year. The total length of the season is given by the difference between the cessation and onset dates. For the simulations, the calculation is performed separately for each year in each ensemble member. For each day of each year (and of each ensemble member, in the case of the simulations), the daily rainfall anomaly to be cumulated is obtained as the rainfall for that day minus the long-term climatological mean daily rainfall using all available years in the observations and all years and ensemble members in both experiments, AMV+ and AMV-, in the models."

- statistical significance: 'To test whether the change in a given quantity is statistically significant we apply the parametric test for differences of means under independence (Wilks, 2019)' Please describe more precisely what kind of parametric test you have used. I assume you have used a t-test. I am wondering if this test is applicable for extreme values (e.g. Figure 14), because the populations must be normally distributed?

Thank you for your comment. We have indeed used a t-test. To clarify this, we have changed the sentence to: "To test whether the change in a given quantity is statistically significant we apply the parametric t-test for differences of means under independence, assuming a Gaussian distribution for the samples (Wilks, 2019)"

Regarding the concern raised by the reviewer on the use of this t-test for Fig. 14, we note that we are evaluating frequencies (number of days) and not the actual extreme values. The latter would indeed be better fitted with a Generalized Pareto Distribution (Wilks, 2019). The number of days are more normally distributed, even more so because the values we are testing are averages over the 10 years of simulation (we are only assuming each ensemble member as an independent realization and not each year, as explained in lines 187-188 of the manuscript). To show this point further, we have tested for the case of the number of extreme rainy days (Fig. 14) whether the sampling distributions of the 10-year means are inconsistent with a Gaussian distribution using the Kolmogorof-Smirnov goodness of fit test with the variant of having fitted the parameters of the distribution (also known as Lilliefors test, see Wilks 2019). Fig. R1 shows that the Gaussian distribution is in general not inconsistent with the samples of 10-year averages of the extreme rainy days in regions where there is some summer rainfall (typically above 1 mm/day of mean JAS rainfall). This suggests that for most of the areas where significant differences in the number of extreme rainy days between AMV+ and AMV- experiments are shown in Fig. 14 of the manuscript, the assumption of Gaussian distribution used in the t-test is not inconsistent with the distribution of the samples used to evaluate it.

[Figure]

Figure R1: Average JAS rainfall (shaded, mm/day) and regions where the distribution of the samples of 10-year averaged extreme rainy days is inconsistent with a normal distribution (dotted areas) for either the AMV+ and / or the AMV- ensembles. For each grid point, the Lilliefors test has been applied separately for the AMV+ and AMV- ensembles samples of 10-year averaged extreme rainy days. Dots mark where the null hypothesis of normal distribution is rejected at p=0.05 for any of the AMV+ and the AMV- ensembles.

- The color bar (magnitude and units) in some plots might be wrong. I think they do not fit to the caption. Please have a look e.g. at Figure 2 and Figure 5. The caption of Figure 2 says mm/day, but in the figure it says days. I have the impression that also the magnitude of the color-bar is not correct.

Many thanks for this remark. We apologize for these mistakes. We have revised carefully the units and magnitudes in the figures and made the following changes:

* Fig 1: we have added the units for the climatological contours in plots c and d (same units as the corresponding anomalies).

* Fig 2: we have corrected the units in the color bar to "mm/day". Magnitudes were already right in the former version. The reviewer might have thought magnitudes were wrong because of the values of rainfall biases shown in Fig. 3ab. However, we note that in Fig. 3ab we only plot the averaged biases over the Sahel box (10ºE-10ºW, 10ºN-20ºN). Local values might reach biases as strong as -6 mm/day (for instance the coast of Guinea Bissau in the IPSL-CM6A-LR model). To avoid confusion to the readers, we have changed the sentence in lines 194-197 from "All in all, the models provide too dry conditions over the Sahel ranging from deficits of 0.3 mm/day for the CNRM-CM6-1 model with the DCPP-C protocol to 1.8 mm/day for the EC-Earth3 in DCPP-C and ECMWF-IFS-LR in PRIMAVERA ones, which roughly represent between 8 and 60 % of average rainfall over the Sahel from MSWEP (Fig. 3ab)." to "Averaged over the Sahel box, the models provide too dry conditions ranging from deficits of 0.3 mm/day for the CNRM-CM6-1 model with the DCPP-C protocol to 1.8 mm/day for the EC-Earth3 in DCPP-C and ECMWF-IFS-LR in PRIMAVERA ones, which roughly represent between 8 and 60 % of average rainfall over the Sahel from MSWEP (Fig. 3ab)."

* Fig 5: we have corrected the units in the color bar and in the caption to "mm/day". We have also corrected the caption for the contours which are the climatological values of the intensity of rainfall and not the number of rainfall days, as it was mistakenly written in the former version.

* Fig 6: we have corrected in the caption the eastern boundary of the westernmost Sahel to 10ºW. When presenting the westernmost Sahel region in the text (line 180) we have expanded the description from "(purple box in Fig. 1cd)" to: "(purple box in Fig. 1cd, taken as the region 17ºW-10ºW, 10ºN-20ºN, after removal of the area west of the line connecting the points 17ºW-12ºN and 15ºW-10ºN).

* Fig 9: we have corrected in the caption the units of the anomalies shaded to mm/day.

- wondering if it might be useful to also compare the precipitation PDFs of the observations and simulations for some key areas.

Following this suggestion, we present in Fig. R2 the PDF of daily precipitation averaged over the Sahel box (10ºW-10ºE, 10ºN-20ºN) for models and observations. Compared to MSWEP, models tend to overestimate (underestimate) the number of days with daily rainfall amounts below (above) 2-3 mm/day averaged over the Sahel. This is particularly problematic for EC-Earth3 model in DCPP-C protocol and the ECMWF-IFS-LR and ECMWF-IFS-HR models in the PRIMAVERA protocol. These model biases are yet beyond our estimate of observational uncertainty: CHIRPS data provides a PDF with a tendency to even higher values of rainfall than MSWEP (Fig. R2h).

The model biases in the PDF of daily rainfall over the Sahel box are consistent with the analysis of the bias in mean rainfall intensity presented in the manuscript: Figs. 3a and 5 already suggested that, beyond the observational uncertainty, models tend to show too weak mean intensity of rainfall over the Sahel, especially for EC-Earth3 model in DCPP-C protocol and the ECMWF-IFS-LR and ECMWF-IFS-HR models in the PRIMAVERA protocol. This is, at the grid point level, when it rains, it tends to rain smaller amounts than in the observations.

Given that the article is already quite long, especially the section on model biases (figures 2 to 6), and that part of the information provided by showing PDFs is conveyed in the analysis of the mean intensity of rainfall, we prefer to leave the PDF analysis out of the manuscript.

[Figure]

Figure R2: Histograms of daily rainfall in the JAS season averaged over the Sahel box (10ºW-10ºE, 10ºN-20ºN) for the models (orange bars, plots a to g) and MSWEP (black lines in all plots) and CHIRPS (orange bars in plot h). Horizontal axis shows daily rainfall values (mm/day) and vertical axis shows the number of days per JAS season. All days have been used for the calculation (no temporal average has been applied). The simulations following the PRIMAVERA protocol are marked as blue in the model name labels.

Further modifications done to the manuscript:
- We have updated the acknowledgment section to include thanks to the anonymous reviewer and to add an additional source of funding.

References cited:

-Liebmann, B., Blade, I., Kiladis, G. N., Carvalho, L. M. V., Senay, G. B., Allured, D., Leroux, S., and Funk, C.: Seasonality of African Precipitation from 1996 to 2009, Journal of Climate, 25, 4304–4322, https://doi.org/10.1175/JCLI-D-11-00157.1, 2012.

-Wilks, D. S.: Statistical Methods in the Atmospheric Sciences, 4th Edition, Elsevier, Amsterdam, https://doi.org/10.1016/C2017-0-03921-6, 2019.

---

## Author Comment (AC2)

**Review of "Impact of AMV on rainfall intensity distribution and timing of the West African Monsoon in DCPP-C-like simulations" by Mohino et al.**

Authors have analysed the impact of AMV on West African precipitation (monsoon season length, intensity and spatial changes) in a small set of model simulations of different characteristics using different SST forcing following 2 AMV protocols, model physics, and different resolutions. Authors found a coherent impact of AMV on WAM precipitation among the simulations. Under AMV+, the increased precipitation change found in the southern part of WAM has been explained through the increase in wet days and extreme events relative to the northern counterpart.

**Major:** the statistical analysis performed is overall convincing, although I do have some doubts about the robustness of the conclusions because the chosen simulations do not allow a clear understanding if differences are arising because the model resolution, the DCPP-C or PRIMAVERA protocol, because the model physics (being usually very large for precipitation) or AMV phases. The simulations used in this study run under different model version and this makes the comparison very complicated. Additionally, is it not really clear to me the premise of the study. Also, I did not fully get what is the added value of using these simulations instead of CMIPs or other type of simulations which perform AMV decently. I guess that the most important stuff here is the bias assessment due to AMV phases that could help to understand overall north-south or east-west precip biases in WAM. I invite authors to revise the manuscript trying to calibrate better the focus of the study. Also, mechanisms for explaining the biases are not investigated at all, and I think that would be nice to know more about where the differences among models come from.

We would like to begin by thanking the reviewer for the time taken in reading and commenting our manuscript. In the following, we give answer to all the concerns raised using red to distinguish our answers. We start by answering the points highlighted in this first major comment.

Regarding the focus of the study, previous works show an impact of AMV on seasonal rainfall amounts. However, in this study we hypothesize that AMV might also have an impact on the intra-seasonal characteristics of rainfall (timing and distribution of rainfall). To evaluate this hypothesis, we use a modeling approach and show that the models consistently show such impacts. Prior to the evaluation of this hypothesis, we want to evaluate the performance of the models in simulating rainfall and its intraseasonal characteristics. For this reason, we start by analyzing the biases of the models in their simulation of the distribution of rainfall and timing of the monsoon. We do not intend to explain the origin of these biases, as the experimental protocol is not designed for such purpose.

Regarding the added value of the simulations we use, the DCPP-C and PRIMAVERA protocols allow for a very consistent forcing in the sensitivity experiments performed with the models. We are forcing upon them the exact same pattern, namely the observed AMV pattern, with the same seasonal timing, while avoiding other sources of variability that are present in CMIP historical or preindustrial simulations. In preindustrial and historical simulations, patterns of AMV change from one model to the next (*e.g.*, Martin et al. 2014) and differ from the observed one, so it is not simple to disentangle if differences between the model's results are due to the differences between the SST anomalies or between the response to the SST anomalies (or both).

Regarding the differences in the simulated responses, we acknowledge that the use of the two protocols could introduce non-linearities when comparing model's responses. Nevertheless, we do show and discuss that the changes in protocol are of second order importance to the differences

shown among models. Indeed, as explained below, the same CNRM-CM6-1 model has been run under both protocols, so the comparison between both runs allows for a clear evaluation of the impact of the protocol. As can be seen in Figs. 6, 7, 8, 9, 11 and 12, the differences between the results of CNRM-CM6-1 model under both protocols are much smaller than those shown with respect to any other model. This provides evidence pointing at differences in results by model coming from the model's response and not from the protocol followed. As for the differences in resolution, we also discuss in the last part of the manuscript that there is no clear relation between the resolution of the models and the simulated impacts. Finally, we do show in Fig. 3cd and discuss in sections 3.2 and 4 that there is evidence suggesting the biases might affect the way the models simulate the amplitude of the changes.

Regarding the robustness of the conclusions, we show that, despite the differences among models, there is a common response to a positive AMV phase consistent in a general increase in JAS seasonal rainfall amounts that overlays the climatological values, with higher changes in the southern Sahel, typically close to 10ºN, and weaker ones to the north. The ones on the north are mainly related to an increase in the number of rainy days due to the enhancement of all types of rainfall events, moderate, heavy, and extreme. The ones of the south are better explained by an increase in the mean intensity of rainfall, with an increase in the number of heavy and extreme rainfall events, while the number of moderate events changes slightly and even decreases over some locations. In addition, most models suggest a lengthening of the monsoon season in response to a positive AMV phase, principally due to a later demise. The changes in the timing are much clearer in the westernmost Sahel, with all models showing an earlier onset, later cessation, and longer monsoon length.

To make these points clearer, we have revised the text and made the following changes:

- We have modified the abstract to make clearer our focus (see next comment). We clarify in the abstract that "Here we seek to explore these impacts [whether and how AMV affects the distribution of rainfall or the timing of the West African Monsoon] …". Our aim (exploring the impacts of AMV on the timing and rainfall distribution of the West African monsoon) is now clearly stated in the title, abstract, introduction and summary and discussion sections.

- We have added the relevance of a consistent pattern across simulations in the introduction by expanding sentence in lines 72-75 to: "Here we take this approach and analyze the output of experiments run with global coupled climate models in which the North Atlantic SSTs are flux-corrected to provide idealized AMV-like anomalies, as proposed in component C of the Decadal Climate Prediction Project (DCPP-C) (Boer et al., 2016), which allows for a very consistent AMV forcing across model simulations."

- We have modified the beginning of section 3.1 (biases) to emphasize our aim is in evaluating the impacts but that we start by analyzing model's biases. The sentence in line 192-194 has been expanded to: "Before evaluating the impacts of AMV, we analyze model's biases in representing the timing and distribution of rainfall. Biases are estimated by first averaging the AMV+ and AMV-experiments and then subtracting the observational estimate."

**Abstract:**

Ln 3: better to state immediately what kind of dataset you are using (e.g. CMIPs… or other…). It is not clear what models are you talking about now because later on you introduce DCPPs…

Ln 7:"… models…" once again, not clear what kind of coupled models are you talking about.

Ln 9: "land mass"… I would change in "over land".

Ln 15: "stronger negative biases" relative to what? Observations? Please clarify.

Thank you for these suggestions. We have taken them into account and modified the abstract to:

"Previous studies agree on an impact of the Atlantic Multidecadal Variability (AMV) on total seasonal rainfall amounts over the Sahel. However, whether and how AMV affects the distribution of rainfall or the timing of the West African Monsoon is not well known. Here we seek to explore these impacts by analyzing daily rainfall outputs from climate model simulations with an idealized AMV forcing imposed in the North Atlantic, which is representative of the observed one. The setup follows a protocol largely consistent with the one proposed by the Component C of the Decadal Climate Prediction Project (DCPP-C). We start by evaluating model's performance in simulating precipitation, showing that models underestimate it over the Sahel, where the mean intensity is consistently smaller than observations. Conversely, models overestimate precipitation over the Guinea Coast, where too many rainy days are simulated. In addition, most models underestimate the average length of the rainy season over the Sahel, some due to a too late monsoon onset and others due to a too early cessation. In response to a persistent positive AMV pattern, models show an enhancement in total summer rainfall over continental West Africa, including the Sahel. Under a positive AMV phase, the number of wet days and the intensity of daily rainfall events are also enhanced over the Sahel. The former explains most of the changes in seasonal rainfall in the northern fringe, while the latter is more relevant in the southern region, where higher rainfall anomalies occur. This dominance is connected to the changes in the number of days per type of event: the frequency of both moderate and heavy events increases over the Sahel's northern fringe. Conversely, over the southern limit, it is mostly the frequency of heavy events which is enhanced, affecting the mean rainfall intensity there. Extreme rainfall events are also enhanced over the whole Sahel in response to a positive phase of the AMV. Over the Sahel, models with stronger negative biases in rainfall amounts compared to observations show weaker changes in response to AMV, suggesting systematic biases could affect the simulated responses. The monsoon onset over the Sahel shows no clear response to AMV, while the demise tends to be delayed and the overall length of the monsoon season enhanced between 2 and 5 days with the positive AMV pattern. The effect of AMV on the seasonality of the monsoon is more consistent to the west of 10ºW, with all models showing a statistically significant earlier onset, later demise and enhanced monsoon season with the positive phase of the AMV. Our results suggest a potential for the decadal prediction of changes in the intraseasonal characteristics of rainfall over the Sahel, including the occurrence of extreme events."

Ln 38 "Brazil's Nordeste" North-east Brazil maybe sounds better in this context, because Nordeste usually refers to socio-cultural division. It is up to you.

Thank you for the suggestion. The sentence now reads "It has been shown to modulate the location of the Atlantic Intertropical Convergence Zone (ITCZ), promoting in its positive phase a northward shift of the ITCZ and enhanced rainfall over Amazonia, decreased rainfall over northeast Brazil and

an increased frequency of Atlantic hurricanes (e.g. Knight et al., 2006; Trenberth and Shea, 2006; Zhang and Delworth, 2006; Villamayor et al., 2018a; Hodson et al., 2022)."

Ln 41 "… AMV can also promote". Positive or negative AMV?

Thanks for pointing this out. We have added that it is in response to a positive AMV. The sentence now reads: "Away from the Atlantic, a positive AMV can also promote wetter than average conditions for the Indian Monsoon, warmer than average conditions over northeast Asia, a cooling over the eastern and central tropical Pacific, modifying its inter-annual variability (Ruprich-Robert et al., 2017, 2021; Monerie et al., 2019, 2021; Hodson et al., 2022)."

Ln 47 Again, what AMV phase is promoting enhanced summer seasonal rainfall … of Sahel?

Thanks again for pointing this out. In the new version, we clarify it is the positive phase. The new sentence reads: "There is also a broad consensus, based on observations and modeling studies, that AMV modulates the West African Monsoon, its positive phase promoting enhanced summer seasonal rainfall over the semi-arid area of the Sahel (e.g. Folland et al., 1986; Knight et al., 2006; Zhang and Delworth, 2006; Mohino et al., 2011; Ting et al., 2011; Martin and Thorncroft, 2014; Martin et al., 2014; Villamayor et al., 2018b; Monerie et al., 2019; Hodson et al., 2022) (Fig. 1c)."

**Datasets:**

Ln 113: what do you mean here for ensemble members? How many?

As explained in the manuscript, the experiments consist in 10-yr long simulations with SST restored in the North Atlantic. To account for the uncertainty internal to the climate system, different realizations of the experiment were performed for each model, with each realization (member) differing in the initial conditions. The total number of runs, or ensembles, vary from one model to another and are listed in table 1 ($4^{th}$ column).

To account for this comment, we have modified the first paragraph in section 2.2 by adding the following sentences in line 111: "To account for the uncertainty internal to the climate system, for each model different ensembles members differing in the initial conditions were run (see table 1 for the ensemble size)."

Ln 125: you can include some info about DCPP-C and PRIMAVERA protocol and simulations in the abstract, so it is clear since the beginning on what product your analysis is based on.

Following this comment, and as presented above, we have modified the abstract so now we briefly describe the simulations at the beginning of the abstract ("climate model simulations with an idealized AMV forcing imposed in the North Atlantic, which is representative of the observed one").

Perhaps I have missed it somewhere, but are the simulations running with different SST protocols under piControl or historical set up?

For the DCPP-C protocol, the set-up is consistent with preindustrial conditions. For the PRIMAVERA protocol, as explained in lines 121-123 in the manuscript, the set-up is consistent with mid-Twentieth Century conditions. To emphasize more the DCPP-C pre-industrial set up, we

have modified sentence in lines 106-108 of the manuscript so that it now reads: "Here we make use of the AMV experiments in DCPP-C which are 10 year-long coupled simulations run under pre-industrial external forcing, in which the model's SST are restored in the North Atlantic basin, excluding the Mediterranean Sea, to follow a fixed and idealized pattern of SST anomalies representative of the observed AMV (Boer et al., 2016).". In addition, we have modified lines 121-123 to better detail the PRIMAVERA protocol and now the sentence reads: "However, they differ from the DCPP-C protocol in that they impose twice the anomalous AMV patterns and that the model's setup is based on the control-1950 experiment of HighResMIP (Haarsma et al. 2016), which has fixed forcings representative of the 1950s."

**Metrics:**

I have a question about the method: would it be easier and more appropriate to count the wet days and then calculating the mean intensity, considering the length of the monsoon season calculating the onset and the withdrawn of the WAM? I think that referring to standard metrics used in the vast literature on monsoons would ease eventual comparison with other studies without introducing new metrics. How would the number of wet days you have introduced compare with the common monsoon season length?

See Sultan and Janicot 2003, https://journals.ametsoc.org/view/journals/clim/16/21/1520-0442_2003_016_3407_twamdp_2.0.co_2.xml?tab_body=fulltext-display

We are not sure if the suggestion refers to a definition of wet days based on the onset and demise local at each point, which would then be potentially different from one point to another and among simulations and observations. In this case, we argue it is not common to do so. The definition of wet days is usually related to a fixed time frame, either annual or for the summer season (e.g. Sanogo et al. 2015; Diaconescu 2015; DeLongueville 2016; Diakhaté et al. 2019; Diatta et al. 2020; Badji et al. 2022). In addition, a local definition would complicate the interpretation of the results, as AMV can impact the onset and demise in some regions, so a change in the number of rainy days and the derived mean intensity could also be due to the differences in onset and /or demise.

For the above two reasons (comply with usual approach in the definition and avoid to further complicate the interpretation of results) we have chosen to define wet days as days above 1mm of daily cumulated rainfall and consider only a fixed part of the year for this definition. As to what part to use, we decided to focus on the core of the monsoon region, from July to September. Sultan and Janicot (2003) already showed that the monsoon onset date was on average the 24 June. For this reason, we selected the peak season July to September. The results are not affected by our choice of the season, as very similar results are obtained when the analysis is repeated for the extended June to September season (see figures R1 to R3 in this document).

Following this remark, we have clarified the last part of the first paragraph of section 2.3 (metrics) by changing sentence in line 137-138 to: "This season starts just after the average monsoon onset (Sultan and Janicot 2003) and represents the mature phase of the West African Monsoon, when rainfall is well developed in the Sahel (Thorncroft et al., 2011). Conclusions are insensitive to the choice of the start of the summer season, as similar results are obtained when using the extended June to September season (not shown)."

[Figure]

Figure R1: As in Fig. 8 of the manuscript but calculated for the JJAS season instead of JAS.

[Figure]

Figure R2: As in Fig. 9 of the manuscript but calculated for the JJAS season instead of JAS.

[Figure]

Figure R3: As in Fig. 10 of the manuscript but calculated for the JJAS season instead of JAS.

**Statistics:**

I did not really get what statistical test have you used. Would a parametric test be appropriate considering the few data you have?

This same concern was raised by reviewer 1. Indeed, we used a parametric t-test. To help clarify this, we have changed the sentence in lines 184-185 to: "To test whether the change in a given quantity is statistically significant we apply the parametric t-test for differences of means under independence, assuming a Gaussian distribution for the samples (Wilks, 2019)".

To evaluate if the assumption of a Gaussian distribution (as is assumed in the t-test) is valid for our data, we have focused on Fig. 14 and evaluated whether the sampling distributions of the 10-year means are inconsistent with a Gaussian distribution using the Kolmogorof-Smirnov goodness of fit test with the variant of having fitted the parameters of the distribution (also known as Lilliefors test, see Wilks 2019). Fig. R4 shows that the Gaussian distribution is in general not inconsistent with the samples of 10-year averages of the extreme rainy days in regions where there is some summer rainfall (typically above 1 mm/day of mean JAS rainfall). This suggests that for most of the areas where significant differences in the number of extreme rainy days between AMV+ and AMV- experiments are shown in Fig. 14 of the manuscript, the assumption of Gaussian distribution used in the t-test is not inconsistent with the distribution of the samples used to evaluate it.

[Figure]

Figure R4: Average JAS rainfall (shaded, mm/day) and regions where the distribution of the samples of 10-year averaged extreme rainy days is inconsistent with a normal distribution (dotted areas) for either the AMV+ and / or the AMV- ensembles. For each grid point, the Lilliefors test has been applied separately for the AMV+ and AMV- ensembles samples of 10-year averaged extreme rainy days. Dots mark where the null hypothesis of normal distribution is rejected at p=0.05 for any of the AMV+ and the AMV- ensembles.

**Results:**

Figure 2: Label panel b and d: How CNRM-CM6-1 black and blue differ? Is it just a different simulation with different protocol but with the same model? And also, how do EC-Earth3 and 3P-HR differ? Just resolution or also something else?

As explained in the caption of Fig. 2, the models whose name is drawn in blue follow the PRIMAVERA protocol. The difference between Fig.2b and Fig.2d is thus just the protocol followed, the model is the same.

EC-Earth3 is the standard model version used for CMIP6 (Döscher et al. 2022). It has an horizontal resolution of T255. The EC-Earth3P-HR is a higher resolution version of the EC-Earth3 developed under PRIMAVERA project (Haarsma et al. 2020). Both share the atmospheric IFS cycle 36r4 NEMO 3.6 ocean and LIM3 sea-ice components. They are, however, run at different resolutions. In addition, EC-Earth3P-HR branched from EC-Earth3 at an early stage of development. Döscher et al. 2022 gives a summary of the main differences between both versions, which include a different treatment of stratospheric aerosols and vegetation.

Following this comment from the reviewer, we have added an explanation in section 2.2 (Simulations) to explain more clearly that the CNRM-CM6-1 model was run under both protocols. The sentence "To evaluate these potential non-linearities we analyze the simulations done by the CNRM-CM6-1 model, which has been run under both protocols (table 1)" has been added to the end of line 127.

What is the east-west bias in CNRM due to?

We speculate with a possible answer to this question in the following comment on the SST biases of the models.

Ln 200: it would be great also to show SST north-south SST bias in all datasets (in the supplementary for example).

In Fig. R5 we show the SST biases presented by the models. As with most current state-of-the-art models, the ones analyzed in the manuscript also present prominent warm biases in the southeastern tropical Atlantic. As explained in the manuscript (lines 200-205), there is a body of work showing that these biases tend to shift the ITCZ southwards, promoting a dry bias over the Sahel and a wet bias over the Gulf of Guinea.

Interestingly, next to the West African coast north of 15ºN, the CNRM-CM6-1 under both protocols presents a warm bias above 1ºC, which is not present in any other of the analyzed models. We speculate this subtropical North Atlantic warm bias could be contributing to the wet biases in the westernmost coast of West Africa in this model. A warming in the subtropical North Atlantic could indeed be enhancing the supply of moisture inland through the low-level westerly jet, promoting a more unstable atmosphere (Pu and Cook, 2012), potentially offsetting the effect of the tropical biases (Giannini et al. 2013), especially in the westernmost Sahel.

As explained previously, our main interest in presenting the model's biases is not to explain them but to evaluate the performance of models in simulating rainfall and its intraseasonal characteristics over West Africa. We have therefore avoided overloading the paper with explanations of the potential origin of the shown biases. Neither the experimental set up nor the analyzes performed are aimed at explaining the biases of the coupled models.

[Figure]

Figure R5: Biases (shaded, units are ºC) in mean JAS sea surface temperatures. Simulations under PRIMAVERA protocol are marked as blue in the model name labels. Biases are estimated as

differences between the model's SSTs and the ERSSTv4 averaged between 1901 and 2013. For reference, contours mark observed climatology (in ºC).

Ln 200 on: discussing biases in tropical precipitations among CMIP generations you can refer to Fiedler et al., 2020: Fiedler, S., Crueger, T., D'Agostino, R., Peters, K., Becker, T., Leutwyler, D., ... & Stevens, B. (2020). *Monthly Weather Review, 148*(9), 3653-3680.

Thank you for this suggestion. We have modified the sentence in lines 200-201 in the former manuscript and it now reads: The north-to-south rainfall biases, which are still a common feature of CMIP-6 models (Fiedler et al. 2020), are consistent with the warm biases of SST simulated by all models in the southeastern tropical Atlantic, which reach values well over 2ºC (not shown)."

**Discussion:**

Ln: 430 – 436: It is nice to see some proposed explanations for these biases. It would be great also to see some figures about, related to the simulations you have used, especially on soil moisture, otherwise it is just speculation.

Thank you for this suggestion. In the commented text we were not referring to model biases. This paragraph is linking changes in the delayed demise of the monsoon season in response to a positive phase of AMV with the increase in the amount of rainfall fallen. Following this suggestion, we show in Fig. R6 the differences in total soil moisture content (the mass per unit area summed over all soil layers of water in all phases) between AMV+ and AMV- simulations. The figure is consistent with the proposed mechanism of enhanced rainfall due to a positive phase of AMV providing enhanced soil moisture content which, in turn, would lead to a later demise through land-atmosphere interactions.

We have therefore modified the sentence in lines 431-433 of the manuscript to: "As much of the rainfall falling over the Sahel comes from local recycling (Nieto et al., 2006), this higher consistency in the demise date could be related to the increased soil moisture in models that follows an enhanced rainfall season in response to a positive AMV phase (not shown)." We have moved this sentence to section 3.5 (Impacts of AMV on the timing of the monsoon season).

[Figure]

Figure R6: Difference in mean JAS total soil moisture content (the mass per unit area summed over all soil layers of water in all phases) between AMV+ and AMV- experiments (shaded, kg/m$^2$). For simulations under PRIMAVERA protocol (marked as blue in the model name labels) only half the anomalous values are shown. Regions where differences are not statistically significant (p<0.05) are dotted.

I also suggest to shorten and make clearer the main findings in this section.

Thank you for this suggestion. We have modified the last section so that we now use bullet points to mark the main findings of the manuscript, making a clear separation between the summary of the results and the discussion. On the other hand, we have removed part of the discussion that was mixed with the result summary. In particular, we have removed the sentences in lines 415-420 of the manuscript ("We note these biases …. estimates shown in this work.") and also those in lines

427-429 ("This enhancement in the amount and intensity of rainfall … flood risk (Tazen et al., 2019; Elagib et al., 2021)." The part on the soil moisture as an explanation for the more consistent delay in the demise of the monsoon in models after a positive AMV phase has been moved to section 3.5 (Impacts of AMV on the timing of the monsoon season).

From your analysis, is it possible to understand if the type of storms is expected to change, e.g., increasing the formation of Mesoscale Convective Systems (MCS)? See here: Fitzpatrick, R. G., Parker, D. J., Marsham, J. H., Rowell, D. P., Guichard, F. M., Taylor, C. M., ... & Tucker, S. (2020). What drives the intensification of mesoscale convective systems over the West African Sahel under climate change?. *Journal of Climate, 33*(8), 3151-3172.

Unfortunately, our analysis does not provide an answer to the reviewer's question. We cannot tell how the type of storms might be affected by AMV. Our main source of data is daily cumulated rainfall. We have not analyzed hourly or 3-hourly data of rainfall nor of OLR to follow and classify individual storms. In addition, despite the somewhat high resolution, none of the analyzed models are storm-resolving, so they can have difficulties to accurately model storms (Marsham et al. 2013).

Further modifications:
- We corrected a typo in the name of one of the authors: Donnat → Donat

References cited in this reply that were not in the manuscript:
-Giannini, A., Salack, S., Lodoun, T., Ali, A., Gaye, A. T., & Ndiaye, O. (2013). A unifying view of climate change in the Sahel linking intra-seasonal, interannual and longer time scales. *Environmental Research Letters, 8*(2), 024010.
-Marsham, J. H., Dixon, N. S., Garcia-Carreras, L., Lister, G. M., Parker, D. J., Knippertz, P., & Birch, C. E. (2013). The role of moist convection in the West African monsoon system: Insights from continental-scale convection-permitting simulations. *Geophysical Research Letters, 40*(9), 1843-1849.
-Pu, B., & Cook, K. H. (2012). Role of the West African westerly jet in Sahel rainfall variations. *Journal of Climate, 25*(8), 2880-2896.